# RAST-MoE-RL: A Regime-Aware Spatio-Temporal MoE Framework for Deep Reinforcement Learning in Ride-Hailing

**Yuhan Tang** [* 1]  **Kangxin Cui** [* 2]  **Jung Ho Park** [* 3]  **Yibo Zhao** [* 1]  **Xuan Jiang** [1]  **Haoze He** [4]  **Jiangbo Yu** [5]
**Haris N. Koutsopoulos** [6]  **Jinhua Zhao** [1]

## Abstract

Ride-hailing platforms must balance passenger waiting times with overall system efficiency under highly uncertain supply–demand conditions. Adaptive delayed matching, which controls the holding intervals for batched sets of requests and vehicles, exposes an inherent trade-off between matching and pickup delays. The non-stationary nature of request arrivals and dynamic congestion exposes a key limitation of existing methods, which rely on shallow encoders that cannot capture these spatio-temporal regime shifts. We introduce the Regime-Aware Spatio-Temporal Mixture-of-Experts (RAST-MoE) framework, which formalizes adaptive delayed matching as a regime-aware Markov Decision Process and equips RL agents with a self-attention Mixture-of-Experts encoder, letting different experts specialize automatically across operational regimes while keeping per-sample computation modest. On real-world TNC trajectory data from San Francisco, our 12M-parameter model reduces average matching delay by 10% and pickup delay by 15%, trains stably without reward hacking, generalizes zero-shot to two additional cities, and exhibits clear expert specialization across demand regimes. These results demonstrate the value of MoE-enhanced RL for large-scale decision-making tasks with complex spatio-temporal dynamics.

## 1. Introduction

The optimization of ride-hailing platforms has traditionally been formulated as combinatorial optimization (CO) problems (Alonso-Mora et al., 2017; Duan et al., 2020; Zhang et al., 2020), but suffered from scalability issues with growing fleet and passenger batch sizes. RL offers a strong alternative, since the system evolves over long horizons under stochastic demand and supply, satisfying the Markov property (Mazyavkina et al., 2021). In this formulation, one effective strategy to improve the system performance is to control the matching time interval for more efficient assignment of batched supply and demand (Qin et al., 2021). As this requires close coordination among the autonomous agents in the environment, a centralized RL agent is a natural design choice (Ning & Xie, 2024; Riley et al., 2021), but several challenges still remain to be deployed in real world effectively. First, most prior works assume stationary congestion, limiting transferability of trained policies to environments with varying traffic conditions. Second, long-horizon training can be unstable, as agents exploit loopholes by repeatedly selecting default or "do-nothing" actions rather than improving service (Amodei et al., 2016). Third, standard policy networks often lack the expressiveness to capture complex spatiotemporal patterns, leading to poor adaptability across different supply–demand regimes and times of day (Zhang et al., 2018).

To address these issues, we propose leveraging Mixture-of-Experts (MoE) architectures as compact, regime-aware encoders. MoE models enable a divide-and-conquer representation, where specialized experts handle different operating regimes and a gating mechanism adaptively selects among them (Jacobs et al., 1991). This improves representation capacity while keeping computation tractable, since only a subset of experts is activated per decision step. This enables policies to adapt to non-stationarity with good sample efficiency. We base our study on prior works on RL for ride-hailing, congestion modeling, and MoE architectures in RL; a more detailed discussion is deferred to Section 2.

Our work makes three main contributions: (1) We develop a physics-informed, congestion-aware environment that formalizes adaptive delayed matching as a **Regime-Aware**

---

[*]Equal contribution  [1]Massachusetts Institute of Technology [2]Tongji University [3]University of California, Berkeley [4]Carnegie Mellon University [5]McGill University [6]Northeastern University. Correspondence to: Xuan Jiang <xuanj@mit.edu>, Haoze He <haozeh@cs.cmu.edu>.

*Proceedings of the 43$^{rd}$ International Conference on Machine Learning*, Seoul, South Korea. PMLR 306, 2026. Copyright 2026 by the author(s).

**Spatio-Temporal MDP (RAST-MDP).** A macroscopic surrogate for travel times preserves the essential density–speed feedback while remaining efficient enough for millions of RL rollouts. (2) We design a reward scheme that combines incremental matching and pickup costs with adaptive service constraints. An online multiplier adjusts penalties based on service-quality violations, preventing reward hacking strategies and stabilizing training over long horizons. (3) We develop RAST-MoE, a compact MoE-based encoder with only 12M parameters that achieves consistent improvements across algorithms. Despite its modest size, it outperforms strong baselines on real-world TNC data, improving total reward by 13% and reducing both matching and pickup delays.

## 2. Related Work

There have been several well-studied formulations in the literature utilizing RL to control the matching intervals for better system-level efficiency of the ride-hailing platforms. Notably, RL has been applied to ride-hailing matching and dispatch through zone-based centralized agents with reward decomposition (Qin et al., 2021), multi-agent formulations with trajectory replay (Ke et al., 2022), and centralized PPO with potential-based reward shaping (Bao et al., 2025). These past works address challenges of sparse rewards and scalability, but approximate travel times with static cost matrices, overlooking congestion-related dynamics which is time-variant. As a result, their policies offer limited evidence of generalization across heterogeneous demand conditions.

Replicating time-variant congestion is a necessary tool for testing the real-world robustness of RL policies. While GNN- and GCN-based surrogates can approximate traffic dynamics (Zhao et al., 2020; Narayanan et al., 2023; Pham et al., 2025), they often require retraining under new conditions and lack interpretability. By contrast, MFD preserves the physics of density–speed relations while remaining computationally efficient (Geroliminis & Daganzo, 2008; Beojone & Geroliminis, 2023; Huang et al., 2024; Zhang et al., 2024), which makes it a strong choice for scalable RL environments.

Mixture-of-Experts (MoE) architectures scale model capacity by routing inputs to specialized experts (Shazeer et al., 2017). This divide-and-conquer design allows different experts to capture distinct regimes of the state–action space. This makes MoE very appropriate to apply for ride-hailing platform environment since the environment inherently experiences clear cyclic congestion, as well as supply and demand patterns. In reinforcement learning, MoE has been shown to improve sample efficiency and stability in multimodal or non-stationary settings, for example through mixtures of deterministic experts (Osa et al., 2023), probabilistic

policy ensembles (Ren et al., 2021), and soft-gated modules that maintain balanced expert utilization (Obando Ceron et al., 2024; Puigcerver et al., 2024). Willi et al. (2024) further highlight MoE's ability to adapt under distribution shifts, underscoring its value for dynamic environments. Beyond RL control, spatio-temporal MoE encoders such as ST-MoE-BERT (He et al., 2024) demonstrate strong performance in modeling human mobility, demonstrating its capabilities in large-scale transportation domains. However, their application to large-scale mobility control remains largely unexplored. Our work closes this gap by combining congestion-aware surrogates with a regime-aware MoE encoder tailored to ride-hailing.

## 3. Challenges of Adaptive Delayed Matching in Ride-Hailing

Adaptive delayed matching in ride-hailing is a decision problem between immediate or delayed matching of drivers with batched passenger requests for improved matching. This creates a trade-off between matching delay (request–assignment) and pickup delay (assignment–pickup), managed under evolving supply–demand and congestion patterns. Empirically, cyclic demand and congestion patterns form data distributions with clustered regimes (e.g., morning peaks, off-peak). A monolithic encoder must interpolate across such patterns, often yielding suboptimal actions (Ren et al., 2021), while MoE mitigates this by routing to specialized experts—validated in our expert utilization analysis (Sec. 6.3). Such challenges are not unique to ride-hailing: delayed matching and dispatch occur broadly in transportation systems, logistics, and large-scale on-demand services, making this problem of general importance and such a setting motivates more expressive representations that can adapt across regimes and stable policy optimization across millions of simulator rollouts.

Recent advances in large language model (LLM) training provide useful building blocks for this problem. Mixture-of-Experts (MoE) architectures scale capacity efficiently by sparsely activating only a few experts per input, providing three advantages directly relevant to ride-hailing control: (i) training efficiency—higher effective capacity under the same FLOPs, essential when RL requires millions of rollouts; (ii) parameter efficiency—even modest-sized MoE models can capture complex structure, important under memory limits; and (iii) specialization—different experts naturally adapt to distinct supply–demand regimes such as peak congestion versus off-peak sparsity. Meanwhile, RL methods such as PPO (Schulman et al., 2017) stabilize long-horizon policy improvement under noisy, high-variance rewards, which makes it a good basis for adaptive delayed matching. Combining MoE with PPO therefore makes a suitable choice for ridehailing applications, as MoE

offers regime-aware high-capacity representations at low cost, while PPO provides training stabilities.

While we acknowledge the success of MoE and PPO in their native domains, directly integrating the two for ride-hailing application is inappropriate unless the formulation explicitly exposes regime heterogeneity. PPO typically employs relatively simple MLPs for its actor and critic networks, and injecting non-stationarity may degrade training efficiency or stability (Moalla et al., 2024). In such cases, combining MoE with PPO becomes appealing: MoE allows specialization of experts to distinct regimes, while PPO provides a known and stable on-policy update mechanism. This integration thus strikes a balance between expressive, regime-aware representation and controlled policy improvement. We therefore formalize adaptive delayed matching as a RAST-MDP, injected with supply–demand imbalance. Then, we empirically test multiple MoE parameters to tune the architecture for best performance. From this point, section 4 specifies RAST-MDP in detail (state, action, transition, and an anti-hacking reward), including a robust physics-informed surrogate for network travel times. Section 5 then presents RAST-MoE-RL, which is a PPO-based learner with a regime-aware spatio-temporal MoE encoder, designed to solve RAST-MDPs efficiently and stably at the city scale.

# 4. Regime-Aware Spatio-Temporal MDP (RAST-MDP)

We introduce the Regime-Aware Spatio-Temporal MDP (RAST-MDP) as a principled formulation of adaptive delayed matching in ride-hailing. RAST-MDP is designed to faithfully capture the complexities of large-scale urban operations by reconstructing key components of the MDP: (i) spatio-temporal states that reflect heterogeneous demand–supply regimes, (ii) a combinatorial action space for zone-wise batching, (iii) an adaptive reward design with safeguards against pathological strategies, and (iv) a physics-informed travel-time surrogate.

Together, these components yield a realistic yet scalable environment where RL policies can be trained robustly and evaluated fairly. We now describe each element in detail.

## 4.1. State, Action, and Transition.

The state vector $s_t$ denotes the system state at time $t$, aggregating supply, demand, and temporal statistics across zones:

$$s_t = \left[\tau_t,\ \rho_t,\ n_p^{(1:N)},\ n_d^{(1:N)},\ \lambda^{(1:N)},\ \mu^{(1:N)}\right] \in \mathbb{R}^{d_s},$$

where $\tau_t$ is the time of day, $\rho_t$ the remaining horizon, $n_p^{(i)}$ and $n_d^{(i)}$ the unmatched passengers and idle drivers in zone $i$, and $\lambda^{(i)}$ and $\mu^{(i)}$ the stochastic arrival rates of passengers

and drivers. The state space captures the heterogeneity of supply–demand dynamics across space and time.

Actions are binary at the zone level: $a_t^{(i)}=1$ means "match now in zone $i$" and $a_t^{(i)}=0$ means "hold." The joint action space is therefore $\mathcal{A} = \{0,1\}^N$, which grows exponentially with the number of zones. This combinatorial structure makes the task challenging and highlights the need for expressive function approximation that can exploit spatiotemporal structure.

The system evolves under a stochastic kernel $s_{t+1} \sim P(\cdot \mid s_t, a_t)$, which executes matching in the chosen zones, continues waiting for held requests, advances drivers' ongoing trips and repositioning, and introduces new passenger and driver arrivals. Pickup travel times are determined by a physics-informed surrogate (§4.3) that preserves the feedback between density and speed while remaining efficient enough for millions of RL rollouts. Episodes begin with a short warm-up period to seed realistic queues, and temporal domain randomization is applied to improve policy robustness.

## 4.2. Anti-Hacking Reward Design

A critical challenge in applying RL to ride-hailing is reward design: naive time-weighted formulations are brittle and invite reward hacking. For example, over-penalizing pickup delay can drive the policy to reject distant passengers or hold requests indefinitely, while over-penalizing batching delay collapses to myopic "match-immediately" behavior. To avoid such pathologies, we design instantaneous reward $r_t$ around incremental costs with adaptive safeguards:

$$r_t = -\left[c_m\,\Delta W_t^{\text{match}} + c_p(t)\,\Delta W_t^{\text{pickup}}\right] - \lambda_t\,(g(s_t,a_t)-\alpha).$$

Here $\Delta W_t^{\text{match}}$ and $\Delta W_t^{\text{pickup}}$ are the marginal increases in matching and pickup waits (hours), weighted by $c_m$ and $c_p(t)$. The pickup weight $c_p(t)$ is modulated from the surrogate model (§4.3), encouraging batching in free flow but discouraging it under heavy load. The soft constraint $g(s_t,a_t)\leq\alpha$, where $g(s_t,a_t)$ measures service-quality violations (we use the fraction of late matches or pickups) and $\alpha$ is a tolerance level, is enforced by an adaptive multiplier updated online,

$$\lambda_{t+1} \ \leftarrow\ \left[\lambda_t + \xi\,(g(s_t,a_t) - \alpha)\right]_+,$$

so that penalties increase when violations exceed the tolerance and relax otherwise. In our study, we allowed at most 5% of requests to exceed a delay threshold, which represents realistic service-quality thresholds used by major Transportation Network Companies (TNCs). Please refer to Appendix H for performance sensitivity to $\alpha$. Therefore, the algorithm learns the batching–service balance instead of relying on a fixed hand-tuned ratio (e.g., the 4:1 weights in

Qin et al. (2021)). Note that $\xi$ acts as a learning rate for $\lambda$, and that, in a real-world environment with finite demand and cyclic fluctuations, the adaptive multiplier should converge even in the presence of demand spikes. The convergence plot and further intuition of $\lambda$ are illustrated in Appendix L.

This formulation ensures that (i) rewards reflect marginal operational impact, (ii) collapse strategies such as indefinite holding or selective matching are unattractive, and (iii) the batching–pickup trade-off adapts naturally to evolving supply–demand regimes.

### 4.3. Robust, Physics-Informed Travel Time Surrogate Model

Accurate travel-time feedback is essential for capturing the batching–pickup trade-off, since it directly determines downstream pickup durations, driver availability, and the true impact of matching decisions on service quality. Microscopic simulators are too computationally expensive for millions of RL rollouts, while constant-speed heuristics flatten spatiotemporal variability and distort long-horizon credit assignment. Prior work has shown that physics-informed coordinated-flow representations offer high-fidelity approximations of urban link speeds while remaining computationally lightweight (Yu & Hyland, 2025), making them well-suited for RL settings that require millions of rollouts. To reconcile realism with scalability, we build a physics-informed surrogate that compresses microscopic signals into zone–hour macrostates via macroscopic fundamental diagrams (MFDs).

Our construction proceeds in three steps: (i) *Zone–hour speeds.* Aggregate per-edge flow/density into zone-level macrostates and enforce the MFD relation $q = kv$, yielding zone–hour speeds $v_z^{(h)} = q_z^{(h)}/k_z^{(h)}$. (ii) *Static routes.* Run a one-time shortest-path search to obtain a fixed path set $\{\mathcal{P}_r\}_{r\in\mathcal{R}}$ covering all OD pairs. (iii) *Hourly OD times.* For each hour $h$, compute travel times by accumulating edge lengths with zone-uniform speeds: $\tau_r^{(h)} = \sum_{e\in\mathcal{P}_r} \frac{\ell_e}{v_{z(e)}^{(h)}}, h = 0, \ldots, 23$, where $\ell_e$ is edge length and $z(e)$ maps edge $e$ to its zone.

At rollout time, travel-time queries are answered in $O(1)$ by reading $\tau_r^{(h_t)}$ for the current hour $h_t$, with no online microsimulation or ad hoc rescaling. Because $v_z^{(h)} = q_z^{(h)}/k_z^{(h)}$ is MFD-consistent, the surrogate retains essential congestion signals—including capacity drops at high density—so the agent experiences the correct batching vs. pickup structure. Aggregation at the zone/hour scale smooths microscopic noise while preserving peak/off-peak regime shifts, improving long-horizon credit assignment. All heavy routing is performed offline, so the approach scales naturally to city-scale networks and millions of RL steps. Further construction details appear in Appendix A.

## 5. Solving RAST-MDPs by RAST-MoE-RL

Building on the RAST-MDP formulation in Section 4, this section specifies a compact actor–critic learner equipped with a regime-aware spatio-temporal mixture-of-experts encoder. We refer to this framework as **RAST-MoE-RL** (Regime-Aware Spatio-Temporal MoE for Reinforcement Learning). Figure 1 summarizes the overall architecture: observations are embedded into tokens, contextualized by a lightweight Transformer, routed through MoE experts, and then shared by actor and critic heads for training. PPO is adopted as the primary trainer, and the same plug-in encoder remains compatible with A2C, ACER, and GRPO, yielding consistent gains.

### 5.1. Policy Optimization (PPO as Primary Trainer)

At decision epoch $t$, the policy factorizes over zones as a product of Bernoulli heads,

$$\pi_\theta(a_t \mid s_t) = \prod_{i=1}^{N} \text{Bernoulli}\big(a_t^{(i)} \mid \sigma(\ell_\theta^{(i)}(s_t))\big),$$

where $a_t^{(i)}{=}1$ denotes "match-now" in zone $i$. Note that even while the policy factorizes over zones, it still achieves global coordination by operating on global pooled state vectors that contain information about all zones; hence, an action in zone $i$ depends on the state of its neighboring zones as well. A centralized state-value function $V_\theta(s_t)$ serves as a variance-reducing baseline, and advantages $\hat{A}_t$ are computed via generalized advantage estimation (GAE). Training maximizes the clipped surrogate augmented with entropy regularization and a value penalty,

$$\mathcal{L}_{\text{PPO}}(\theta) = \mathbb{E}_t\left[\min\left(r_t(\theta)\hat{A}_t, \text{clip}(r_t(\theta), 1-\epsilon, 1+\epsilon)\hat{A}_t\right)\right.$$
$$\left. - \beta\mathcal{H}(\pi_\theta(\cdot|s_t)) + c_v\left(V_\theta(s_t) - R_t\right)^2\right]$$

where $r_t(\theta) = \pi_\theta(a_t \mid s_t)/\pi_{\theta_{\text{old}}}(a_t \mid s_t)$ and $R_t$ is the return. The actor and critic share a single feature extractor, which is replaced by the proposed RAST-MoE so that both heads consume regime-aware spatio-temporal embeddings. This substitution preserves PPO rollouts and the objective while increasing representational capacity at approximately fixed per-sample compute.

### 5.2. RAST-MoE: Regime-Aware Spatio-Temporal MoE Feature Extractor

Ride-hailing control requires reasoning over heterogeneous spatiotemporal states (city grid, demand/supply flows) and a large discrete action set (delayed-matching decisions). Shallow MLP encoders fail to capture long-range spatial dependencies, while dense Transformers are computationally

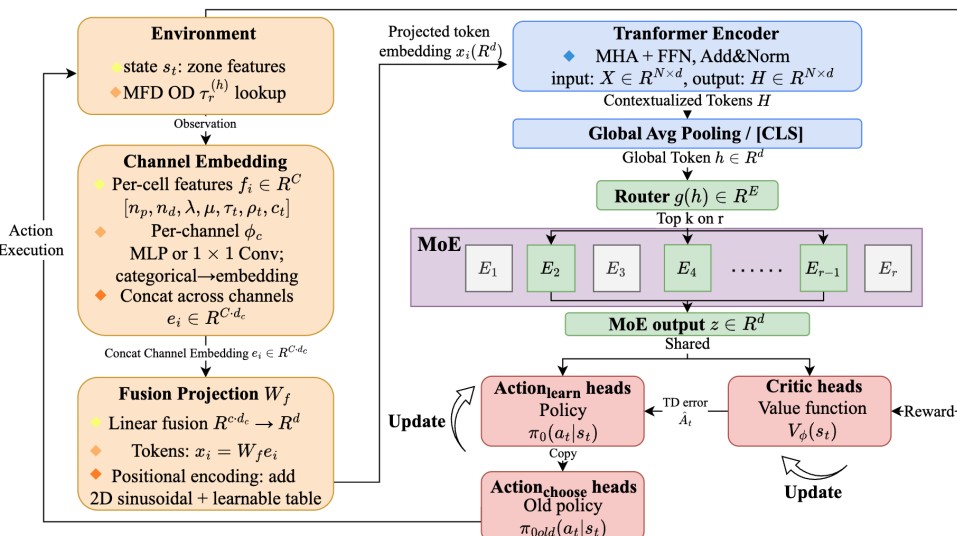

*Figure 1.* RAST-MoE-RL learner for RAST-MDPs. A shared encoder is replaced by a regime-aware spatio-temporal MoE; actor and critic heads remain lightweight.

expensive. We therefore introduce RAST-MoE, a Regime-Aware Spatio-Temporal Mixture-of-Experts encoder that balances expressivity and efficiency. RAST-MoE replaces the shared feature extractor in PPO; the actor and critic heads remain lightweight linear projections, ensuring that performance gains can be attributed to improved representation learning.

At each decision epoch $t$, the city is partitioned into $H \times W$ cells ($N=HW$). For each cell $i$, we observe $(n_p^{(i)}, n_d^{(i)}, \lambda^{(i)}, \mu^{(i)})$ (passengers, drivers, arrival rate, driver arrival rate), with optional temporal features $c_t \in \mathbb{R}^{d_c}$. Each channel is embedded and fused as

$$x_i = W_f \left[\phi_p(n_p^{(i)}); \phi_d(n_d^{(i)}); \phi_\lambda(\lambda^{(i)}); \phi_\mu(\mu^{(i)})\right] \in \mathbb{R}^d,$$

where $\phi_\bullet$ are small MLPs and $W_f$ projects to hidden size $d$. Spatial layout is encoded by sinusoidal positional embeddings plus a learnable location table, and temporal covariates are broadcast after a linear map. The token matrix $\tilde{X} = [x_1, \ldots, x_N]^\top \in \mathbb{R}^{N \times d}$ is then processed with a lightweight self-attention encoder, and global average pooling produces a compact state representation $h \in \mathbb{R}^d$.

To capture heterogeneous operating regimes, the router operates on the pooled global state $h$, rather than token-level embeddings. This choice improves computational efficiency, stabilizes training, and reflects the fact that ride-hailing control requires global spatio-temporal reasoning rather than fine-grained token decisions. A two-layer MLP with GELU activation produces gating logits $g(h) \in \mathbb{R}^E$. We adopt sparse top-$K$ routing: only the $K$ experts with the highest logits are activated, and their outputs are aggregated with

normalized softmax weights,

$$\text{MoE}(h) = \sum_{e \in \mathcal{I}} \frac{\exp(g_e(h))}{\sum_{j \in \mathcal{I}} \exp(g_j(h))} f_e(h),$$

where $\mathcal{I}$ is the set of selected experts. This design follows successful practices in large-scale MoE while ensuring that only a small subset of experts is active per decision step, keeping rollouts efficient.

To prevent routing collapse, we introduce a lightweight load-balancing mechanism that limits excessive concentration on a few experts. Unlike uniform balancing, this design tolerates naturally skewed expert utilization, allowing certain experts to specialize in rare but critical demand regimes—consistent with the inherent imbalance of real ride-hailing systems.

In LLMs, expert balancing has traditionally relied on auxiliary losses (Rajbhandari et al., 2022; Shen et al., 2024; Wei et al., 2024), though recent work such as DeepSeek-V3 shows that lightweight bias terms can achieve similar effects with less interference (Liu et al., 2024). We acknowledge their success in that domain, but note that ride-hailing control presents a different regime structure: supply–demand dynamics are intrinsically imbalanced across time and space—peak-hour congestion and off-peak sparsity call for experts that may be activated at very different frequencies. Imposing strict uniformity can therefore suppress useful specialization and reduce performance. Ablation experiments confirm that both high-frequency and low-frequency experts play critical roles because masking either group degrades performance. Uneven but purposeful specialization is thus not a pathology but an appropriate adaptation to this domain.

The expert-aggregated output $z$ is finally projected into policy and value heads:

$$\pi_\theta(a \mid s_t) = \text{Cat}(W_\pi \sigma(W_z z)), \quad V_\theta(s_t) = w_v^\top \sigma(W_v z).$$

The overall objective is the standard clipped PPO loss with entropy regularization and value loss; no additional auxiliary terms are required.

# 6. Experiments

## 6.1. Experiment Setup

We empirically evaluate our proposed RAST-MoE framework on real-world ride-hailing traces. Our experimental design follows five goals: (i) assess training performance and convergence, (ii) test generalization to unseen demand patterns, (iii) analyze expert specialization and utilization, (iv) examine reward robustness against reward hacking, and (v) verify the necessity of individual architectural components through ablations. Below we summarize the structure of our experiments; detailed figures and results follow in subsequent subsections.

### 6.1.1. DATASETS

We use the official 2019 Annual Reports from the (California Public Utilities Commission , CPUC) Transportation Network Companies (TNC) Data Portal, which provides detailed trip-level records from Uber and Lyft, including pickup/dropoff locations, timestamps, and request outcomes. We focus on San Francisco County, where the pre-COVID ride-hailing market was particularly challenging due to high demand and congestion. According to CPUC statistics, San Francisco hosted on the order of 170,000 completed trips per day, making it one of the densest ride-hailing markets in the United States. We choose 2019 rather than more recent years because post-pandemic recovery has not fully restored the same demand intensity, making 2019 the most representative stress-test environment. We use this open dataset to ensure full reproducibility.

For training and evaluation, we partition the 2019 San Francisco trip data into distinct temporal segments. To promote robustness, the training set pools trips from multiple non-contiguous periods (covering both peak and off-peak demand), while the test set is drawn from disjoint time windows that the model never observes during training. This split ensures that the agent cannot memorize specific demand patterns but must instead learn policies that generalize across heterogeneous regimes.

### 6.1.2. BASELINES

We benchmark **RAST-MoE** under a suite of standard deep RL algorithms to assess robustness across on- and off-policy paradigms. Our primary trainer is **PPO** (Schulman et al.,

2017). For on-policy baselines we include **A2C** (Mnih et al., 2016) and **ACER** (Wang et al., 2017); for off-policy we evaluate **DQN** (Mnih et al., 2015).

**GRPO-style normalization.** Beyond these baselines, we adopt a lightweight Group-Relative Policy Optimization (GRPO) variant originally used in LLM preference optimization (Shao et al., 2024). At each state $s_t$, we draw $K$ candidate actions from the old policy, evaluate their one-step rewards, and form a within-state standardized, bounded score

$$\tilde{r}_{t,k} = \tanh\left(\frac{r_{t,k} - \bar{r}_t}{\sigma_t + \varepsilon}\right), \qquad \bar{r}_t = \frac{1}{K}\sum_{k=1}^{K} r_{t,k},$$

$$\sigma_t = \sqrt{\frac{1}{K}\sum_{k=1}^{K}(r_{t,k} - \bar{r}_t)^2}.$$

The tanh-based normalization bounds large rewards and stabilizes training under highly non-stationary regimes, while trajectory returns and advantages remain computed by standard PPO/GAE; no cross-timestep grouping or ranking losses are introduced. A full description is provided in Appendix B.

## 6.2. Training and Testing Performance

Figure 2 reports training and testing rewards across different algorithms. We observe that PPO already outperforms the ACER baseline in both convergence speed and asymptotic reward, confirming its suitability for this long-horizon problem. Adding the RAST-MoE encoder further improves performance: MoE variants achieve faster training progress and higher final rewards, demonstrating that the additional capacity is effectively utilized rather than wasted. Importantly, these gains also transfer to unseen test scenarios. The configuration with 16 experts and top-4 routing under PPO achieves the best balance, yielding the highest rewards in both training and testing. Concretely, this model improves total reward by 13%, while reducing matching wait by 10% and pickup wait by 15%. GRPO combined with MoE also shows strong results, indicating that our encoder is compatible with multiple RL paradigms. These results suggest that RAST-MoE improves both learning dynamics and generalization beyond standard RL baselines.

## 6.3. Expert Specialization and Utilization

In MoE models for ride-hailing, expert utilization is inherently imbalanced: peak vs. off-peak periods and localized congestion patterns occur with very different frequencies. As a result, some experts are activated far more often than others. The key question is whether these rarely activated experts still provide meaningful contributions or simply waste

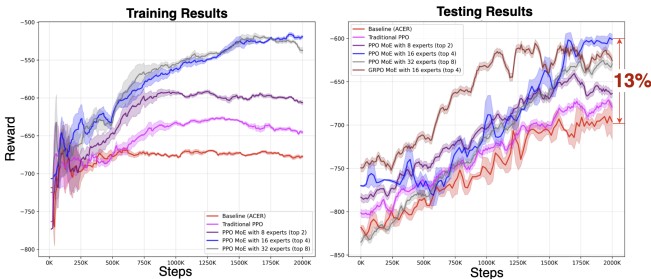

*Figure 2.* Training and Testing Rewards across Baselines and RAST-MoE Variants.

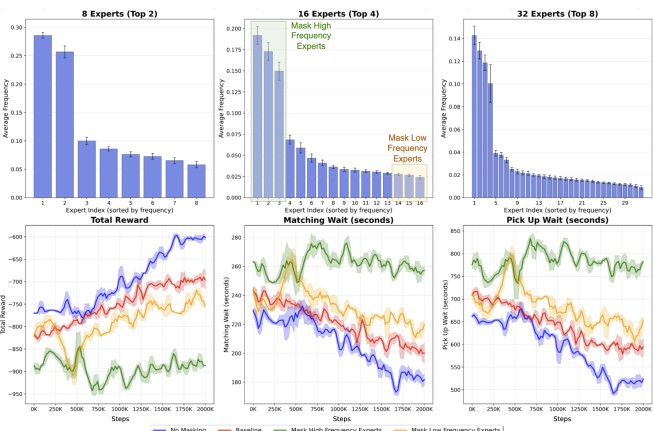

*Figure 3.* Expert utilization and masking results. *Top*: Expert activation distributions under three $(E, K)$ settings with PPO. *Bottom*: Test performance of the 16-expert (top-4) model (the best model as shown in Figure 2) when masking high-frequency (green) or low-frequency (orange) experts, showing total reward, matching wait, and pickup wait.

capacity. To probe this, we examine expert activation frequencies (top row of Figure 3) and find naturally skewed utilization across $(E, K)$ settings, where $E$ is the number of experts and $K$ is the Top-K Routing. We then mask subsets of experts in the 16-expert (top-4) model. As shown in the bottom row, removing either frequent (green) or rare (orange) experts leads to sharp performance drops: rewards decline and both matching and pickup waits stagnate. In contrast, the full model (blue) continues to improve. This demonstrates that both frequent and rare experts encode indispensable regimes, and that flat MoE routing yields purposeful specialization rather than collapse. We also provide an analysis in Appendix I, showing how individual experts correlate with specific demand–supply patterns.

### 6.4. Reward Robustness

Balancing matching wait versus pickup wait is non-trivial: setting coefficients improperly can lead to reward hacking, where the agent exploits the metric rather than improving real performance. In our setting, two pathological behav-

iors dominate: (i) indefinitely holding requests to chase marginal matching improvements, leading to passengers waiting 20–30 minutes or more; or (ii) never holding requests and matching immediately, which quickly exhausts nearby drivers and inflates pickup times. Both strategies can yield superficially high training rewards under certain ratios but are operationally unsustainable.

Figure 4 and Table 1 compare training and testing rewards under different coefficient ratios. While fixed settings such as 1:1 or 4:1 achieve seemingly good training performance and all of the reward goes up in training, they fail to achieve an outstanding result on the test set due to reward hacking. At the extreme, a ratio of 8:1 encourages the policy to always match immediately, driving pickup waits above 25 minutes on average. Conversely, ratios like 1:4 or 1:8 encourage excessive holding: matching waits explode to over 20–30 minutes with very high variance marginal real improvement in pickup times. These outcomes illustrate that fixed coefficients create unstable trade-offs that fail to generalize.

Our adaptive reward (blue) avoids this issue. It matches or exceeds the training performance of fixed settings while maintaining strong generalization in testing. Table 1 further shows that adaptive reward consistently reduces both matching and pickup waits, with lower variance across scenarios. This provides a stable learning signal and prevents reward hacking, ensuring that improvements reflect genuine operational gains rather than metric exploitation.

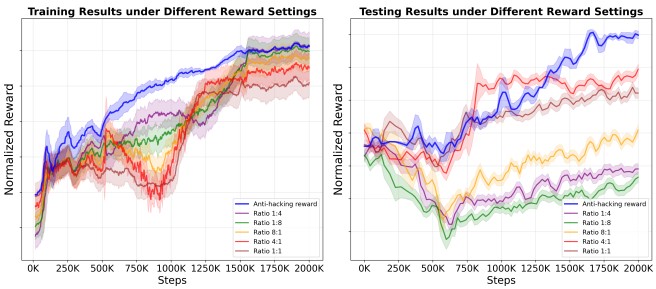

*Figure 4.* Training (left) and testing (right) rewards under different reward coefficient ratios for matching wait vs. pickup wait, using PPO with 16 experts (top-4 routing).

### 6.5. Ablation Studies

To better understand the role of model design choices, we conduct a series of ablations. We report matching and pickup waits directly, as they constitute the reward components. The matching wait and pick up wait for different algorithm families, encoder types, parameter count, and MoE routing are shown in Figure 5.

**Moderate-capacity sparse routing achieves the best matching–pickup trade-off.** Across expert-pool sizes and

*Table 1.* Average value and standard deviation of matching wait and pickup wait (seconds) under different reward ratios. Values in red indicate anomalies: either abnormally low matching wait (near zero) or abnormally high matching/pickup wait (over 1300 seconds), which reflect reward hacking behavior.

| Algorithm | Experts (Top-K) | 1:1 | | 4:1 | | 8:1 | | 1:4 | | 1:8 | | Adaptive | |
|---|---|---|---|---|---|---|---|---|---|---|---|---|---|
| | | Match | Pickup | Match | Pickup | Match | Pickup | Match | Pickup | Match | Pickup | Match | Pickup |
| PPO | 8 (Top-2) | 301±37 | 551±95 | 275±34 | 557±88 | 8±36 | 1582±225 | 1433±185 | 522±81 | 2308±845 | 497±83 | 195±40 | 567±44 |
| PPO | 16 (Top-4) | 283±36 | 508±92 | 257±33 | 536±86 | 0±38 | 1746±232 | 1387±178 | 501±84 | 2236±910 | 456±80 | 181±28 | 524±33 |
| PPO | 32 (Top-8) | 277±35 | 519±89 | 249±34 | 548±91 | 0±37 | 1752±243 | 1521±195 | 509±82 | 2786±1056 | 439±78 | 174±29 | 533±41 |
| GRPO | 16 (Top-4) | 296±38 | 515±93 | 265±35 | 543±90 | 2±39 | 1634±211 | 1491±200 | 506±85 | 2545±950 | 440±82 | 189±17 | 528±32 |

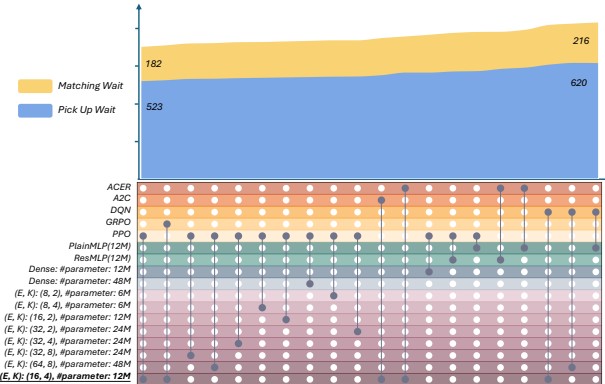

*Figure 5.* **Ablation studies on architectures and training algorithms.** Top: final test-set outcomes (matching wait and pickup wait) across baselines and RAST-MoE variants. Bottom: ablation settings summarizing each configuration—algorithm family, encoder type, and parameter count, and MoE routing $(E, K)$. Entries corresponding to our method are highlighted with bold / underline. The detailed results can be found in Appendix K.

top-$K$ choices, a *mid-scale configuration* realizes the most favorable balance on the test set, whereas further enlarging the pool or excessively sparse activation yields diminishing or adverse returns. The pattern indicates that regime heterogeneity is already captured at *moderate capacity*, while overly large pools tend to increase pickup delays and under-utilize specialization.

**Improvements reflect representation quality rather than optimizer choice or raw parameter counts.** Replacing the shared encoder with MoE consistently enhances outcomes under multiple actor–critic trainers, with *PPO* providing the strongest but not unique gains; analogous benefits appear with GRPO-style normalization, A2C, and ACER. Dense Transformers matched or exceeding the parameter budget do not close the gap, and MLP baselines remain substantially behind, supporting that the advantage stems from regime-aware spatio-temporal representation rather than algorithmic idiosyncrasies or scale alone.

**A compact RAST-MoE attains superior efficiency at small scale.** A lightweight MoE encoder attains lower waits than much larger dense alternatives, and even reduced-capacity variants remain competitive with or surpass dense and MLP baselines. These results highlight an efficiency advantage: expert specialization provides higher effective representational capacity per FLOP, translating into better service-quality trade-offs at modest parameter budgets.

### 6.6. Robustness and Generalization Checks

To further validate our framework, we conduct three additional experiments. First, we benchmark against five non-RL heuristic baselines that share the same environment and MFD surrogate as RAST-MoE-RL. RAST-MoE-RL reduces pickup wait by approximately 33% over the strongest of these (Appendix C). Second, we evaluate cross-city generalization on San Diego City and Alameda County under both native training and zero-shot transfer from San Francisco. The SF-trained policy transfers within 1.5–3% of native performance (Appendix D). Third, we compare our policy against a decentralized execution, where centralized execution outperforms decentralized by 22 s in matching wait and 52 s in pickup wait under the same encoder (Appendix E).

## 7. Conclusion

We introduced RAST-MoE-RL, a regime-aware RL framework for adaptive delayed matching built on a RAST-MDP with explicit zone-wise actions, a physics-informed travel-time surrogate, and an anti-hacking reward. A compact RAST-MoE encoder (12M params) delivers 13% higher reward resulting in 10% and 15% reduction in matching and pickup delays, respectively, while remaining robust to unseen demand regimes. Ablations show a moderate configuration (16 experts, top-4) is best; smaller MoE models still outperform parameter-matched dense Transformers and MLPs; masking either frequent or rare experts degrades performance, indicating meaningful specialization. Overall, regime awareness in both environment and representation is crucial for city-scale control. This approach provides a scalable blueprint for other spatio-temporal resource allocation problems, proving that specialized sub-policies can effectively tackle dynamic complexity.

## Impact Statement

This paper introduces RAST-MoE-RL, a framework that integrates physics-informed constraints and regime-aware representation learning to solve the adaptive delayed matching problem in ride-hailing. Our work advances the application of reinforcement learning to critical urban infrastructure in the following specific ways:

**Operational Robustness and Safety in Physical Systems:** By developing an RL-driven control policy capable of adapting to different operational regimes, the suggested architecture is contributing to the deployment of such policies in highly dynamic, real-world environments. Moreover, by integrating a physics-informed macroscopic fundamental diagram (MFD) surrogate into the training loop, our framework explicitly accounts for the density–speed feedback loops inherent in urban traffic. This addresses a critical "sim-to-real" risk in applying black-box RL to transportation: the potential for policies to exploit simulator inaccuracies and cause gridlock in the real world. Our approach ensures that learned efficiencies are physically plausible, reducing the risks associated with deploying autonomous decision-making in public spaces.

**Efficient and Scalable AI:** As machine learning models grow increasingly resource-intensive, our use of a Mixture-of-Experts (MoE) architecture demonstrates that high-performance control policies do not require massive monolithic networks. With only 12M parameters, RAST-MoE achieves superior performance by activating specialized experts for distinct regimes. This promotes computational efficiency, reducing the cost for training and enabling the deployment of sophisticated control policies on resource-constrained edge computing infrastructure.

**Mitigating Reward Hacking in Service Systems:** A significant risk in automated management is the tendency of RL agents to "game" metrics, for example, by indefinitely holding requests to artificially inflate matching statistics. Our Anti-Hacking Reward Design introduces a mechanism to dynamically penalize service-quality violations. This contributes to the broader effort of designing trustworthy and practical RL systems that remain aligned with intended service-level agreements over long horizons, preventing pathological behaviors that could degrade system reliability.

**Ethical Considerations and Societal Consequences:** While our primary focus is operational efficiency, we explicitly designed the system to mitigate negative externalities often associated with black-box optimization. The "regime-aware" encoder adapts to heterogeneous demand, potentially reducing service gaps in underserved areas, while our service-quality constraints prevent the neglect of individual requests. We acknowledge that algorithmic efficiency must be balanced with labor considerations for gig-economy drivers, and recommend standard monitoring upon deployment. Beyond these operational factors, this work relies solely on anonymized, aggregate traffic flows and does not leverage sensitive personal data or surveillance mechanisms. Consequently, we do not foresee any direct negative societal impacts, dual-use risks, or potential for malicious exploitation associated with this research.

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

## A. Constructing Zone Speeds from LPSim Flow–Density Outputs

### A.1. Surrogate setup and notation

We employ **LPSim** (Jiang et al., 2024a;b;c), a GPU-accelerated, regional-scale, and well-calibrated (Jiang et al., 2025) lane-based microscopic simulator for city-scale networks with time-varying OD demand (Jiang, 2024). LPSim advances vehicles on directed edges (links) indexed by $e$ and records per-vehicle trajectories at 1 Hz (per-second positions). At a fixed reporting interval of width $\Delta t$ (hours), it provides for each edge $e$ its length $\ell_e$ in km and lane count $\lambda_e$ (lanes), together with per-lane density $k_e(t)$ in veh/(lane·km) and per-lane flow $q_e(t)$ in veh/(h·lane) at discrete times $t$; these edge-level time series are aggregated from vehicle presence and link crossings within $[t, t+\Delta t)$. Speeds are not primitives.

We partition the network into transportation analysis zones (TAZ) indexed by $z$. Let $\mathcal{E}_z$ denote the set of edges contained in $z$; we assign each edge to the zone containing its geometry (ties are broken by the edge's tail node). We write $z(e)$ for the unique zone containing edge $e$. The zone's lane-kilometers (our space measure) is

$$L_z \;=\; \sum_{e \in \mathcal{E}_z} \lambda_e\, \ell_e \quad [\text{lane} \cdot \text{km}].$$

### A.2. MFD-consistent aggregation and space-mean speed

In macroscopic fundamental diagram (MFD) terms, the accumulation $N_z(t)$ is the number of vehicles present in zone $z$ at time $t$, and the production $P_z(t)$ is the total vehicle-kilometers traveled per hour within $z$ at time $t$. From edge signals,

$$N_z(t) \;=\; \sum_{e \in \mathcal{E}_z} \lambda_e\, \ell_e\, k_e(t) \;=\; L_z\, k_z(t) \quad [\text{veh}],$$

$$P_z(t) \;=\; \sum_{e \in \mathcal{E}_z} \lambda_e\, q_e(t)\, \ell_e \quad [\text{veh} \cdot \text{km/h}],$$

where the lane-km–weighted macro-averages are

$$k_z(t) = \frac{1}{L_z} \sum_{e \in \mathcal{E}_z} \lambda_e \ell_e\, k_e(t) \quad \big[\text{veh/(lane} \cdot \text{km)}\big],$$

$$q_z(t) = \frac{1}{L_z} \sum_{e \in \mathcal{E}_z} \lambda_e \ell_e\, q_e(t) \quad \big[\text{veh/(h} \cdot \text{lane)}\big].$$

The (zone-level) space-mean speed is distance per vehicle-hour:

$$v_z(t) \;=\; \frac{P_z(t)}{N_z(t)} \;=\; \frac{q_z(t)}{k_z(t)} \quad [\text{km/h}],$$

so the conservation identity $q = k\, v$ holds at the macro level by construction. For numerical stability, we clip very small $k_z(t)$ via $k_z(t) \leftarrow \max\{k_z(t), \epsilon\}$ with negligible $\epsilon$.

*Space–time view over one bin.* Over $[t, t+\Delta t)$, define vehicle-hours and vehicle-kilometers in zone $z$ as

$$A_z \;=\; \sum_{e \in \mathcal{E}_z} \lambda_e \ell_e\, k_e(t)\, \Delta t \quad [\text{veh} \cdot \text{h}],$$

$$D_z \;=\; \sum_{e \in \mathcal{E}_z} \lambda_e q_e(t)\, \ell_e\, \Delta t \quad [\text{veh} \cdot \text{km}],$$

which yield

$$k_z = \frac{A_z}{L_z \Delta t}, \qquad q_z = \frac{D_z}{L_z \Delta t}, \qquad v_z = \frac{D_z}{A_z} = \frac{q_z}{k_z}.$$

*Caution on averaging speeds.* Averaging microscopic $v_e = q_e/k_e$ over edges (even with lane-km weights) generally gives

$$\sum_{e \in \mathcal{E}_z} w_e \frac{q_e}{k_e} \;\neq\; \frac{\sum_{e \in \mathcal{E}_z} w_e q_e}{\sum_{e \in \mathcal{E}_z} w_e k_e},$$

i.e., a time-mean–biased quantity that overestimates space-mean speed in heterogeneous traffic; we therefore always construct $v_z$ from $q_z/k_z$.

### A.3. From zone speeds to an hourly OD travel-time table

We adopt TAZ-uniform hourly speeds. Let $h \in \{0, \ldots, 23\}$ be the hour-of-day index. Aggregating all bins $t$ that fall within hour $h$,

$$A_z^{(h)} = \sum_{t \in h} \sum_{e \in \mathcal{E}_z} \lambda_e \ell_e \, k_e(t) \, \Delta t, \qquad D_z^{(h)} = \sum_{t \in h} \sum_{e \in \mathcal{E}_z} \lambda_e q_e(t) \, \ell_e \, \Delta t,$$

and with $\Delta t_h = \sum_{t \in h} \Delta t$ (typically $\Delta t_h = 1 \, \text{h}$) we set

$$k_z^{(h)} = \frac{A_z^{(h)}}{L_z \, \Delta t_h}, \qquad q_z^{(h)} = \frac{D_z^{(h)}}{L_z \, \Delta t_h}, \qquad v_z^{(h)} = \frac{D_z^{(h)}}{A_z^{(h)}} = \frac{q_z^{(h)}}{k_z^{(h)}} \quad [\text{km/h}].$$

We compute a one-time static route set by Dijkstra (using free-flow speeds or pure lengths as weights), yielding a fixed path $\mathcal{P}_r = (e_1, \ldots, e_{m_r})$ for each origin–destination (OD) pair $r = (o, d)$. Given hourly zone speeds, the OD travel time for hour $h$ is

$$\tau_r^{(h)} = \sum_{e \in \mathcal{P}_r} \frac{\ell_e}{v_{z(e)}^{(h)}} \quad [\text{h}],$$

forming a $24 \times |\mathcal{R}|$ lookup table over the OD set $\mathcal{R}$. During RL rollouts at wall-clock time $t$ with hour index $h_t$, the environment returns $\tau_r^{(h_t)}$ in $O(1)$ without online microsimulation.

## B. Group-Relative Policy Optimization (GRPO): A Lightweight, Practical Variant

GRPO is a relative-reward normalization scheme that compares multiple actions under the same context and leverages their within-context statistics to stabilize policy improvement(Shao et al., 2024). Unlike the LLM preference setting where rewards are noisy and context dependent, our MDP provides well-calibrated scalar rewards; we therefore adopt a minimal GRPO-style component layered atop PPO without altering PPO's surrogate.

### B.1. Algorithmic Description

At each decision state $s_t$, sample $K$ candidate actions $\{a_{t,k}\}_{k=1}^{K} \sim \pi_{\theta_{old}}(\cdot \mid s_t)$ and evaluate their immediate (or truncated) rewards $\{r_{t,k}\}_{k=1}^{K}$. Compute within-state statistics

$$\bar{r}_t = \tfrac{1}{K} \sum_{k=1}^{K} r_{t,k}, \tag{1}$$

$$\sigma_t = \sqrt{\tfrac{1}{K} \sum_{k=1}^{K} (r_{t,k} - \bar{r}_t)^2 + \varepsilon}, \tag{2}$$

$$\tilde{r}_{t,k} = \tanh\left(\tfrac{r_{t,k} - \bar{r}_t}{\sigma_t}\right). \tag{3}$$

The transformation recenters and bounds the per-state samples, mitigating heavy tails and reducing gradient scale sensitivity. **Crucially**, PPO rollouts, return computation, and advantage estimation use the original rewards and standard GAE; the PPO clipped surrogate, value loss, and entropy bonus remain unchanged.

### B.2. Departures from Canonical GRPO

Our implementation differs from canonical GRPO used in LLM preference optimization in three ways:

1. **No cross-timestep grouping.** Grouping is restricted to the $K$ samples at the same state; no contextual keys across time.

2. **No group baselines for advantages.** We do not construct group-centered advantages (no $b_{\mathcal{G}}$); PPO/GAE provides advantages from rollouts.

3. **No pairwise ranking loss.** We add no ranking/contrastive objectives; the PPO surrogate is unchanged.

Canonical GRPO targets noisy, prompt-dependent rewards where relative comparisons are beneficial. In our simulator-driven MDP, rewards are fixed and well-calibrated; cross-context baselines or pairwise order constraints risk diluting absolute-scale information and introduce arbitrary grouping choices. Retaining only within-state normalization keeps the favorable variance/scale properties while preserving unbiased advantage estimation and PPO's simplicity.

## C. Ablation Study: Methods and Configurations

To quantify the contribution of different architectural components and training algorithms, we conduct an extensive ablation study covering MoE variants, dense and MLP encoders, and multiple RL baselines. Performance is evaluated on the held-out test set using two key metrics: average matching wait and average pickup wait. All configurations and results are summarized in Table 2.

In addition to learning-based approaches, we include five heuristic strategies as reference points. Instant Matching assigns each request to the nearest available driver immediately. Notably, this heuristic does not yield zero matching wait because once nearby vehicles are depleted, new arrivals must wait for the next driver to become available, inflating delays. Constant-Interval Batch Matching accumulates requests over a fixed window (20s in our setup) and performs matching at regular intervals. We additionally include three congestion- and regime-aware timing heuristics. They are Best Fixed-Window, which uses a single global holding window tuned on validation data, Time-of-Day Scheduled, which uses a separate holding window for each of four demand periods of the day, and Zone-Level Congestion-Conditioned, which adapts each zone's hold window in real time based on the local MFD congestion index.

*Table 2.* Ablation configurations and outcomes. Rows are ordered by total matching time in ascending order. $(E, K)$ applies to MoE only. Delays are in seconds.

| Encoder / Method | (E,K) | Params | Algorithm | Matching Wait | Pick Up Wait |
|---|---|---|---|---|---|
| MoE | (16,4) | 12M | PPO | 182 | 523 |
| MoE | (16,4) | 12M | GRPO | 185 | 528 |
| MoE | (32,8) | 24M | PPO | 188 | 535 |
| MoE | (64,8) | 48M | PPO | 189 | 536 |
| MoE | (32,4) | 24M | PPO | 191 | 539 |
| MoE | (8,4) | 6M | PPO | 190 | 541 |
| MoE | (16,2) | 12M | PPO | 191 | 543 |
| Dense Transformer | – | 48M | PPO | 192 | 545 |
| MoE | (32,2) | 24M | PPO | 193 | 547 |
| MoE | (8,2) | 6M | PPO | 192 | 548 |
| MoE | (16,4) | 12M | A2C | 197 | 555 |
| MoE | (16,4) | 12M | DPO | 194 | 559 |
| MoE | (16,4) | 12M | ACER | 192 | 569 |
| Dense Transformer | – | 12M | PPO | 201 | 569 |
| GNN | – | 12M | PPO | 208 | 570 |
| ResMLP | – | 12M | PPO | 205 | 575 |
| Plain MLP | – | 12M | PPO | 208 | 577 |
| Dense Transformer | – | 48M | DPO | 204 | 581 |
| ResMLP | – | 12M | ACER | 197 | 589 |
| Plain MLP | – | 12M | DPO | 204 | 589 |
| Plain MLP | – | 12M | ACER | 201 | 596 |
| MoE | (16,4) | 12M | DQN | 211 | 612 |
| Dense Transformer | – | 48M | DQN | 209 | 621 |
| Plain MLP | – | 12M | DQN | 216 | 620 |
| Instant Matching | – | – | Heuristic | 212 | 825 |
| Constant-Interval Batch Matching | – | – | Heuristic | 201 | 842 |
| Zone-Level Congestion-Cond. | – | – | Heuristic | 203 | 784 |
| Time-of-Day Scheduled | – | – | Heuristic | 198 | 802 |
| Best Fixed-Window | – | – | Heuristic | 206 | 814 |

To make sure that the result has statistical significance, we run some representative models using three random seeds (42, 456, 2024) and report mean and standard deviation of matching and pickup waits. We selected (i) the best-performing model, (ii) the strongest dense baseline, (iii) the MLP baseline, and (iv) a secondary RL algorithm, ensuring coverage across architecture families and training paradigms.

*Table 3*. Statistical significance analysis for selected models (3 seeds). Results reported as mean $\pm$ std.

| Model | Algorithm | Matching Wait (s) | Pickup Wait (s) |
|---|---|---|---|
| MoE (16,4), 12M | PPO | 181.58$\pm$0.91 | 523.42$\pm$2.03 |
| Dense Transformer, 48M | PPO | 192.20$\pm$0.87 | 544.63$\pm$1.92 |
| MoE (16,4), 12M | A2C | 196.53$\pm$0.95 | 555.28$\pm$2.16 |
| MLP, 12M | PPO | 208.47$\pm$1.43 | 577.17$\pm$2.35 |

As shown in Table 3, the variance across seeds is small, and the relative ordering of models is preserved in all cases. This confirms that the improvements observed in the main ablations are robust to stochasticity and not driven by single-seed fluctuations.

## D. Cross-City Generalization

We conduct additional experiments on two additional cities (San Diego City (SD) and Alameda County (ALA)) to test the generalization beyond San Francisco. These cities differ from San Francisco in demand distribution, network topology, vehicle density, congestion structure, and spatial demand concentration. SD is sprawling, low-density, and freeway-oriented, with weaker zone-level demand concentration than SF. ALA is polycentric, mixing dense urban cores with sprawling suburban areas, producing a different congestion structure than either SF or SD.

We use two protocols to test increasingly strict notions of generalization. First is native training; the RAST-MoE-RL model is trained and tested on the target city using the same procedure as the SF main results. Second, we test zero-shot transfer. The SF-trained policy is deployed on the target city without any fine-tuning. The agent has never seen the target city's network, zone partition, demand patterns, or congestion profile.

Table 4 reports matching and pickup waits on each target city under each policy, alongside the strongest non-learning heuristic (Zone-Level Congestion-Conditioned), Instant Matching, and the ACER + Plain MLP (12M) RL baseline, which corresponds to the task-specific RL formulation of Qin et al. (2021) matched to our parameter budget.

*Table 4.* Cross-city generalization on SD and ALA. Native denotes train-and-test on the target city; SF→X denotes zero-shot transfer of the SF-trained policy. All values in seconds. Lower is better.

| City | Method | Match Wait (s) | Pickup Wait (s) |
|------|--------|----------------|-----------------|
|      | Instant Matching | 224 | 892 |
|      | Zone-Level Congestion-Cond. | 209 | 832 |
| ALA  | ACER + Plain MLP (12M) | 209 | 641 |
|      | RAST-MoE-RL (SF → ALA) | 196 | 595 |
|      | RAST-MoE-RL (ALA → ALA) | **191** | **578** |
|      | Instant Matching | 220 | 947 |
|      | Zone-Level Congestion-Cond. | 218 | 873 |
| SD   | ACER + Plain MLP (12M) | 215 | 692 |
|      | RAST-MoE-RL (SF → SD) | 205 | 626 |
|      | RAST-MoE-RL (SD → SD) | **208** | **617** |

Under native training, RAST-MoE-RL reduces pickup wait by approximately 30–35% relative to the strongest non-learning heuristic and by 7–10% relative to the parameter-matched ACER + Plain MLP baseline. This consistency confirms that the gains stem from the regime-aware MoE representation rather than SF-specific overfitting. The zero-shot transfer results shows that the policy can generalize to other cities well. Despite never observing the target cities during training, the SF-trained policy achieves pickup waits within 3% of the native policy on ALA and within 1.5% on SD, while still outperforming natively trained ACER baselines.

Note that the SF-trained zero-shot policy on SD obtains slightly lower matching wait (205 s) than the native SD-trained policy (208 s), but higher pickup wait (626 s vs. 617 s). This is because the adaptive Lagrangian reward (Section 4.2) does not optimize a static weighted sum of the two waits. Different training distributions can yield policies that differ by a small trade-off while maintaining similar performance. Both policies remain substantially stronger than the heuristic and plain-MLP RL baselines. Together, these results demonstrate that the RAST-MoE encoder learns generalizable supply–demand–congestion dynamics rather than city-specific patterns.

# E. Centralized vs. Decentralized Formulation

Our main framework adopts a centralized RL formulation. This appendix provides direct empirical evidence that the centralized formulation outperforms a decentralized multi-agent alternative under the same encoder.

We construct a shared-actor multi-agent PPO baseline with decentralized execution. Each zone is treated as an agent that makes its own binary hold/release decision at execution time based only on its local zone observation plus a small set of shared global summaries. To keep the comparison fair, all zone agents share the same actor parameters, since they solve the same type of local decision problem. The MDP transition dynamics, the anti-hacking reward (Section 4.2), the action semantics (binary hold/release per zone), and the evaluation protocol are identical to the centralized setting, so the comparison isolates exactly the centralized-vs-decentralized modeling choice. Table 5 reports matching and pickup waits under each formulation, evaluated with both the Transformer encoder and the proposed RAST-MoE encoder.

*Table 5.* Centralized vs. decentralized execution under matched encoders. All values in seconds. Lower is better. The first two rows are existing results from Section 5.2, and the last two rows are the added decentralized comparison.

| Method | Execution | Match Wait (s) | Pickup Wait (s) |
|---|---|---|---|
| Centralized PPO + Transformer | Centralized | 201 | 569 |
| Centralized PPO + RAST-MoE | Centralized | **182** | **523** |
| Shared-actor MA-PPO + Transformer | Decentralized | 217 | 581 |
| Shared-actor MA-PPO + RAST-MoE | Decentralized | 204 | 575 |

Centralized execution consistently outperforms decentralized execution under the same encoder. With RAST-MoE, the gap is 22 s in matching wait and 52 s in pickup wait. This is expected as a centralized agent can jointly reason over which subset of zones to release in a given epoch, whereas decentralized agents acting independently from local observations cannot fully coordinate this combinatorial release decision, even when sharing actor parameters. RAST-MoE improves both formulations, but the benefit is far larger under centralized execution (pickup drops 46 s centralized vs. 6 s decentralized), confirming that the regime-aware MoE representation is most effective when it operates on the complete spatial state.

The results here show that decentralized agents learn reasonable policies, but the best matching/pickup trade-off requires centralized coordination under strongly coupled supply–demand dynamics.

## F. Training Hyperparameters

Table 6 reports the core hyperparameters used to train our model. Unless otherwise noted, defaults follow standard PPO settings; GRPO-style variants share the same base configuration except where indicated in Appendix B. All experiments were implemented with the Adam optimizer.

*Table 6.* Core hyperparameters for our model

| Parameter | Our Model | Description |
|---|---|---|
| Learning Rate | $2 \times 10^{-4}$ | Optimizer learning rate |
| Batch Size | 512 | Batch size per update |
| n_steps | 2048 | Environment steps per update |
| n_epochs | 5 | Training epochs per update |
| Clipping $\epsilon$ | 0.2 | PPO clipping parameter |
| Clip Range VF | 0.2 | Value function clipping |
| Entropy Coefficient | 0.01 | Entropy regularization weight |
| Value Coefficient | 0.7 | Value loss weight |
| GAE $\lambda$ | 0.95 | Generalized Advantage Estimation parameter |
| Discount Factor $\gamma$ | 0.99 | Reward discount factor |
| Max Grad Norm | 0.4 | Gradient clipping threshold |
| Optimizer | Adam | Optimizer type |

## G. Evaluation of Fixed Reward Weights

The relative weighting between matching delay and pickup delay determines how strongly the reward function favors immediate assignment versus batching, and different fixed-weight choices can lead to distinct trade-offs across demand regimes. To provide a more comprehensive assessment of fixed-weight reward designs, and to better demonstrate the behavior of our adaptive reward mechanism, we extend our analysis to include additional moderate settings under the same training configuration.

**Experimental setup.**    All the hyperparameters, rollout settings, and PPO configurations were the same. Each model was trained for the same number of updates, and evaluated on the same held-out demand regimes.

**Results.**    Table 7 summarizes matching and pickup waits. The adaptive multiplier continues to outperform all fixed weights, achieving consistently lower matching and pickup waits with reduced variance.

*Table 7.* Average matching wait and pickup wait (seconds) under different reward ratios for **PPO, 16 experts (Top-4)**. Values in red indicate anomalies (either abnormally low matching wait near zero or abnormally high matching/pickup wait over 1300 seconds), reflecting reward-hacking behavior. We report means and standard deviations across runs. The 1:1, 4:1, 8:1, 1:4, 1:8, and Adaptive rows are copied from Table 1; 2:1 and 3:1 are newly added moderate fixed-weight baselines.

| Reward Ratio | Matching Wait (s) | Pickup Wait (s) |
|---|---|---|
| 1:8 | 2236±910 | 456±80 |
| 1:4 | 1387±178 | 501±84 |
| 1:1 | 283±36 | 508±92 |
| 2:1 | 291±31 | 511±84 |
| 3:1 | 278±29 | 532±81 |
| 4:1 | 257±33 | 536±86 |
| 8:1 | 0±38 | 1746±232 |
| Adaptive (ours) | **181±28** | **524±33** |

These additional results demonstrate that even when compared against standard industry-style fixed weights, the adaptive reward consistently produces a more stable and higher-performing batching–pickup trade-off.

## H. Sensitivity Analysis of Service-Quality Violation Tolerances

The tolerance parameter $\alpha$ controls the allowable fraction of service-quality violations (e.g., late matches or pickups) in the adaptive reward. A smaller $\alpha$ enforces stricter service guarantees, while a larger $\alpha$ permits more batching flexibility. Prior transportation and ride-hailing studies typically adopt a tolerance of approximately 5% (Zhou et al., 2019; Qin et al., 2022), which serves as our default setting in the main paper.

To evaluate the robustness of our adaptive penalty mechanism and to illustrate how the learned policies behave under different service-level requirements, we extend our analysis using three tolerance levels:

$$\alpha \in \{2.5\%,\ 5\%,\ 7.5\%\}.$$

**Experimental setup.** All experiments use the same PPO configuration, hyperparameters, and rollout settings as in the main results. For each $\alpha$, we train the model for the same number of updates and evaluate it on identical held-out demand regimes to ensure comparability.

**Results.** Table 8 reports matching and pickup waits under the three tolerance levels. Across the full range of $\alpha$, performance remains highly consistent: matching waits vary by less than 7% and pickup waits by less than 2%. The adaptive multiplier $\lambda_t$ adjusts smoothly under all settings without inducing instability or oscillation. These results confirm that platform operators may flexibly choose an appropriate $\alpha$ based on their desired service-level agreements, with 5% representing a reasonable and commonly used baseline.

*Table 8.* Average value and standard deviation of matching wait and pickup wait (seconds) under different service-quality violation tolerances $\alpha$ for **PPO, 16 experts (Top-4)**.

| Tolerance $\alpha$ | Matching Wait (s) | Pickup Wait (s) |
|---|---|---|
| 2.5% | 188±38 | 538±44 |
| 5.0% | 181±28 | 524±33 |
| 7.5% | 193±41 | 531±36 |

The adaptive reward exhibits strong robustness to the choice of $\alpha$: all three settings yield stable learning dynamics and comparable performance across held-out scenarios. This demonstrates that the method can accommodate different SLA requirements while maintaining its efficiency and generalization benefits.

# I. Intuition for Expert–Regime Correspondence

Although Mixture-of-Experts (MoE) models are often considered hard to interpret, we carried out some exploratory analysis to better understand expert utilization in our setting. Focusing on the best-performing configuration (16 experts, top-4 routing), we observe that different experts are activated in ways that correspond to meaningful supply–demand regimes in San Francisco:

- **Peak-onset specialization.** Expert 16 shows high activation (over 25%) during the onset of the evening peak (5–6pm), when supply–demand imbalance in downtown begins to emerge, but is rarely used outside peak hours.

- **High-frequency experts.** Experts 1–3 are broadly activated across most periods (15–20%), suggesting they capture general, recurring patterns that are essential throughout the day.

- **Peak-dominated experts.** Experts 11–12 are more active in peak hours (12–15% during 7–9am and 5–7pm), and much less active off-peak.

- **Off-peak experts.** Experts 7–8 appear more frequently in mid-day and late-night off-peak periods (around 10%), but contribute less than 2% in peaks.

These observations suggest that MoE routing discovers sub-policies that align with real-world traffic regimes: general experts, peak-specific experts, and off-peak experts. While exploratory, this analysis indicates that even infrequent experts encode non-trivial behaviors, strengthening the value of regime-aware specialization.

## J. Expert–Regime Correspondence Through Activation Heatmaps

Although established literature in Mixture of Experts suggests that experts rarely specialize in such clear, human-interpretable categories and the interpretability is beyond our scope, we provide an exploratory qualitative analysis to reveal interesting patterns in how the router utilizes experts throughout the day. Figure 6 visualizes expert activation frequencies over the 24-hour cycle for three representative MoE configurations (8, 16, and 32 experts).

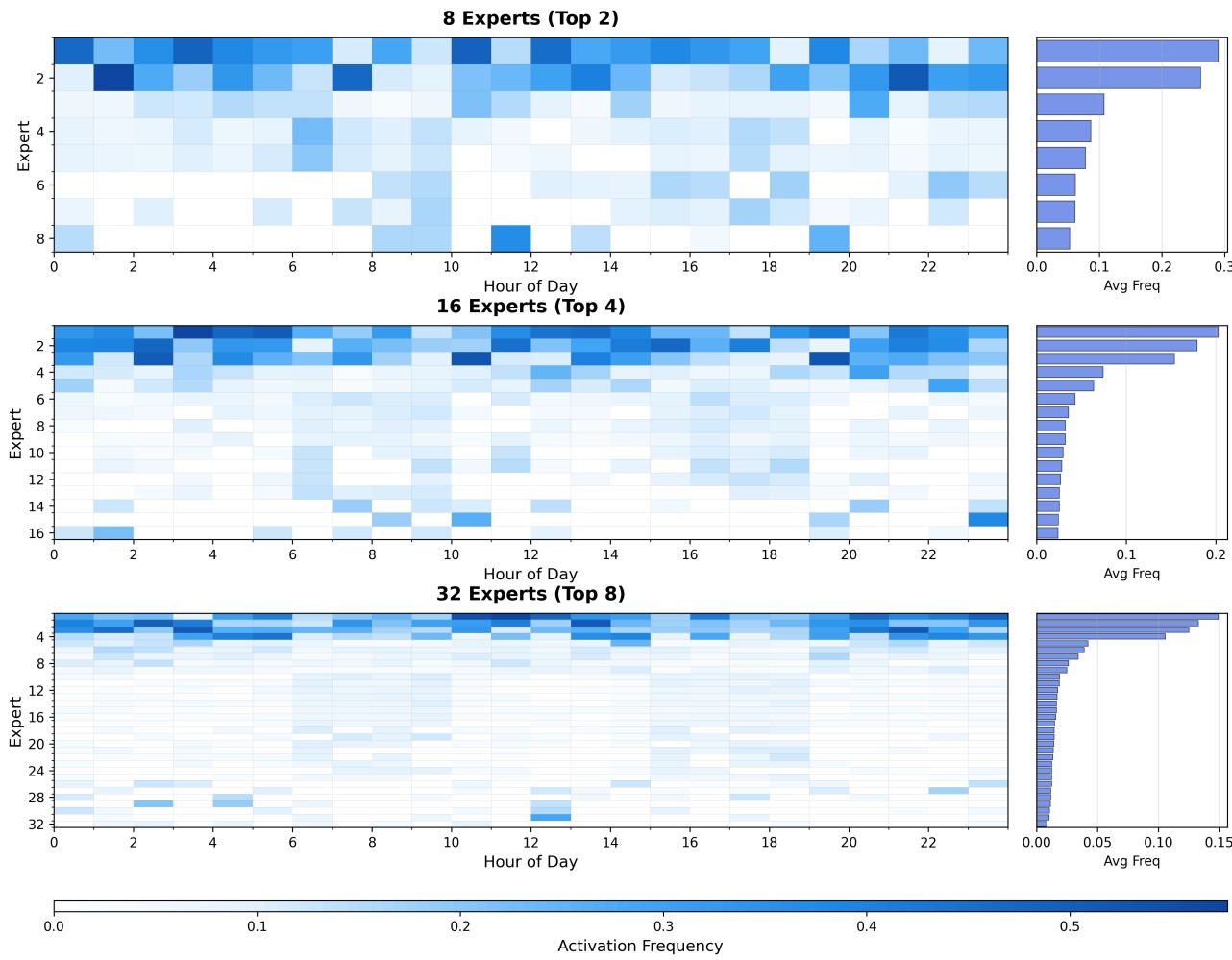

*Figure 6.* Expert activation frequencies over the 24-hour cycle under different MoE configurations (8, 16, and 32 experts). Activation patterns reveal regime-aware specialization, with peak-hour periods exhibiting more evenly distributed routing and off-peak periods handled by a smaller subset of experts.

Figure 6 reveal that the router learns distinct temporal responsibilities for different experts:

- High-frequency experts are activated consistently throughout the day. These experts capture general, city-wide mobility structures that are relevant across all supply–demand regimes.

- Medium-frequency experts exhibit pronounced activation during the morning and evening peak periods (approximately 6–10 AM and 3–7 PM, corresponding to the 6–10th and 15–19th hour of day). These intervals are characterized by strong supply–demand imbalance and elevated congestion. During such periods, the router spreads activation more evenly across multiple experts. This is visually reflected by noticeably darker colors for medium-frequency experts in the peak-hour columns of the heatmaps, indicating higher activation rates compared with off-peak hours. These patterns suggest that peak periods are sufficiently complex that no single expert can dominate and the medium-frequency experts learn how to deal with peak hours.

- Low-frequency experts specialize in other rare or off-peak regimes, including midday and late-night periods. Although activated less frequently, masking experiments (Fig. 5) show that removing these experts leads to noticeable degradation, demonstrating that rare regimes still carry important dynamics.

A noteworthy pattern is that peak-hour activation is more evenly distributed across experts. This indicates that the router decomposes highly heterogeneous and non-stationary peak conditions into multiple expert subspaces, allowing the model to represent congested regimes more flexibly. In contrast, off-peak periods are dominated by a small subset of experts, implying that a few experts are sufficient to handle low-demand or low-congestion dynamics. The heatmap analysis provides direct evidence that the proposed MoE encoder exhibits meaningful regime-aware routing, with different experts specializing in specific supply–demand–congestion patterns across the day.

## K. Robustness Evaluation under Global OD Perturbations and Incident Shocks

To evaluate deviations from hourly conditions (e.g., accidents, weather disruptions, localized bottlenecks), we conducted two new sensitivity-related experiments. We introduced out-of-distribution perturbations for test purposes.

**Scenario A: Global Perturbation (Weather/Citywide Shocks).** We scale all the OD travel times by a global factor $\eta_h \sim \text{Uniform}(0.8, 1.2)$ to capture the citywide slowdowns or surges caused by the possible bad weathers or demand fluctuations.

**Scenario B: Localized Incident Shocks (Accidents/Road Closures).** We sample "accident corridors" of 5–12 edges from the Open Street Map road network and apply a penalty $\alpha \sim \text{Uniform}(1.5, 3.0)$ to OD pairs whose routes go through these edges, creating highly localized congestion spikes beyond the surrogate's nominal range.

Table 9 summarizes the relative degradation in matching and pickup waits. Therefore, the learned RAST–MoE policy does not overfit to static congestion patterns. We have added these robustness experiments to the appendix of the revised manuscript.

*Table 9.* Robustness evaluation under global OD perturbations and incident shocks. Metrics report average matching wait and pickup wait (lower is better). Absolute and percentage changes are relative to the base scenario.

| Model | Base Scenario | | Global Perturbation | | Incident Shock | |
|---|---|---|---|---|---|---|
| | Match Wait (s) | Pickup Wait (s) | Match $\Delta$ | Pickup $\Delta$ | Match $\Delta$ | Pickup $\Delta$ |
| ACER Baseline | 201 | 596 | +12 (6.0%) | +42 (7.0%) | +19 (9.5%) | +119 (20.0%) |
| PPO Transformer (48M) | 192 | 545 | +2 (1.0%) | +11 (2.0%) | +17 (8.9%) | +81 (14.9%) |
| **RAST-MoE (E,K)=(16,4)** | **182** | **523** | **+2 (1.1%)** | **+8 (1.5%)** | **+12 (6.6%)** | **+43 (8.2%)** |

## L. Experimental Plot of the Adaptive Multiplier $\lambda$

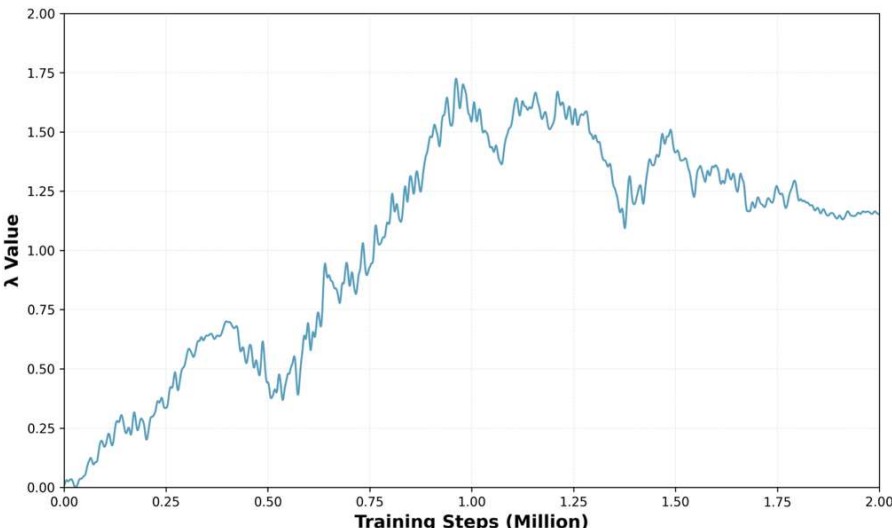

*Figure 7.* Evolution of the adaptive multiplier $\lambda$ over training steps in a representative city-scale experiment. The $x$-axis shows environment steps (in millions), and the $y$-axis shows the current value of $\lambda$. The plot shows $\lambda$ converging to a stable value.

As training proceeds, $\lambda$ starts near zero, steadily increases as the policy initially violates the service-quality constraint, and then stabilizes in a narrow band. We do not observe divergence or large oscillations, even though demand is time-varying over the daily cycle. This matches the intended behavior of the adaptive penalty: when there are frequent violations, $\lambda$ grows and batching becomes more expensive; once the policy learns to keep the violation rate near the tolerance level, $\lambda$ stops drifting and fluctuates around a stable value.

For the intuition behind convergence and stability, recall that the multiplier is updated online as $\lambda_{t+1} = \left[\lambda_t + \xi\left(g_t - \alpha\right)\right]_+$, where $g_t := g(s_t, a_t) \in [0, 1]$ is the service violation, $\alpha$ is the toleration level, and $\xi > 0$ is a step size. Because $g_t \in [0, 1]$, the increment is uniformly bounded:

$$|\lambda_{t+1} - \lambda_t| \leq \xi,$$

so $\lambda$ cannot gain a significant increment in a single update even under demand spikes. For a fixed multiplier $\lambda$, PPO learns a policy $\pi_\lambda$. Denote its long-run average violation rate by

$$\bar{g}(\pi_\lambda) = \limsup_{T \to \infty} \mathbb{E}\left[\frac{1}{T}\sum_{t=1}^{T} g_t\right],$$

where the expectation averages over the daily demand cycle and randomness. Taking expectations of the update gives the approximate drift

$$\mathbb{E}[\lambda_{t+1} - \lambda_t] \approx \xi\left(\bar{g}(\pi_\lambda) - \alpha\right)$$

As long as there exists a feasible policy with $\bar{g}(\pi) \leq \alpha$ and the learned violation rate $\bar{g}(\pi_\lambda)$ decreases as $\lambda$ increases, then there is a value $\lambda^*$ with $\bar{g}(\pi_{\lambda^*}) \approx \alpha$ where the drift changes sign. This $\lambda^*$ acts as a stable equilibrium of the mean dynamics: for $\lambda < \lambda^*$ the update tends to increase $\lambda$, and for $\lambda > \lambda^*$ it tends to decrease it.

Linearizing the drift near $\lambda^*$ via Taylor approximation gives $\mathbb{E}[\lambda_{t+1} - \lambda_t] \approx \xi a\left(\lambda_t - \lambda^*\right)$, where we define $F(\lambda) := \bar{g}(\pi_\lambda) - \alpha$, $F(\lambda^*) = 0$, and $a = F'(\lambda^*) < 0$. The resulting discrete-time system,

$$\lambda_{t+1} - \lambda^* = (1 + \xi a)(\lambda_t - \lambda^*)$$

is stable if $|1 + \xi a| < 1$. This implies the existence of finite range of moderate step sizes $\xi$, such that $0 < \xi < 2/|a|$, which ensures smooth contraction toward $\lambda^*$. Though the exact range is unknown due to the $a$ term, this implies that an excessively large $\xi$ may lead to oscillation.

The trajectory in Figure 7 is consistent with this intuition: $\lambda$ increases while constraint violations are frequent, then settles into a band around its equilibrium.

