# OpenReview forum: "RAST-MoE-RL: A Regime-Aware Spatio-Temporal MoE Framework for Deep Reinforcement Learning in Ride-Hailing"
_ICML.cc/2026/Conference — ICML 2026 regular_

### Official Review · Reviewer_TP5h · 2026-02-28

**Soundness:** 3
**Presentation:** 1
**Significance:** 3
**Originality:** 3
**Overall Recommendation:** 5
**Confidence:** 3

**Summary:**

Summary:

This paper deals with the problem of adaptive delayed matching in ride-hailing platforms, where the central challenge is to balance passenger waiting times against system-level matching efficiency under non-stationary supply–demand conditions. The paper formalizes this as a Regime-Aware Spatio-Temporal MDP (RAST-MDP) and proposes RAST-MOE-RL, a framework that equips a PPO agent with a compact MOE encoder (of 12M parameters). The framework has three main components:

(1) a physics-informed travel time surrogate based on macroscopic fundamental diagrams that provides congestion aware feedback at low computational cost,

(2) an adaptive anti-hacking reward design with an online Lagrangian multiplier to prevent pathological exploitation of fixed reward coefficients, and

(3) a spatio-temporal MoE encoder with sparse top-K routing that enables different experts to specialize in distinct operational regimes (peak, off-peak, etc.).


When evaluated on real-world Uber trajectory data from San Francisco, the method achieves a 13% improvement in total reward, 10% reduction in matching delay, and 15% reduction in pickup delay relative to strong baselines. Ablation study confirms that gains arise from the MoE representation rather than parameter count or optimizer choice.

**Compliance With Llm Reviewing Policy:**

Affirmed.

**Key Questions For Authors:**

1. Why were the ride-hailing-specific RL methods from the related work (Qin et al., 2021; Ke et al., 2022; Bao et al., 2025) not included as baselines? Even if they require adaptation to your MDP formulation, could the paper clarify which modeling assumptions (state/action definitions, objective, simulator access) prevent a faithful re-implementation?

**Limitations:**

Yes.

**Strengths And Weaknesses:**

Strengths:

S1. Well-motivated MoE integration with demonstrated efficiency gains. The paper makes a principled case for why ride-hailing control requires conditional computation: cyclic demand and congestion patterns form clustered regimes that a monolithic encoder must interpolate across, often yielding suboptimal actions. The MoE encoder directly addresses this by allowing specialized experts to handle distinct operating conditions while keeping per-sample computation tractable via sparse top-K routing.

S2. Adaptive reward design addresses a real and under appreciated failure mode. The anti-hacking reward with an online adaptive multiplier is a strong practical contribution. The paper clearly demonstrates that fixed reward coefficient ratios lead to pathological behaviors: either indefinite holding or immediate matching, both achieving superficially high training rewards but failing catastrophically at test time.

S3: Thorough and well-structured experimental evaluation. The ablation study is comprehensive: it spans multiple RL algorithms (PPO, A2C, ACER, DQN, GRPO), encoder types (MLP, ResMLP, dense Transformer, MoE at various scales), and parameter budgets.

Weaknesses:

W1: Limited novelty: the contribution is primarily incremental. While the combination is sensible and effective, none of the individual components: MoE architectures, PPO, MFD-based travel-time surrogates, or adaptive Lagrangian penalties, are new.

W2: Single-city evaluation limits generalizability claims. All experiments are conducted on San Francisco 2019 Uber data. While this is a dense and challenging market, the paper's claims about scalability and generalization to "city-scale control" remains unsubstantiated.

W3: The "lightweight GRPO variant" proposed in the paper is misleading. The defining contribution of GRPO is the elimination of the critic network entirely, which in this paper's architecture would mean removing one of the two heads that the shared MoE encoder feeds. The paper's variant retains PPO's value function, GAE-based advantage estimation, and the full clipped surrogate, reducing the GRPO component to within-state reward normalization with a tanh squash. What remains is quite far from the spirit of GRPO. Further, the paper does not adequately justify why this particular normalization is preferable to simpler alternatives.

W4: Other weaknesses:
- Please carefully format your paper (the reward in 4.2 bleeds out of the text area) and label your equations (at least the important ones)
- The system figure could use a more explanatory caption
- Some figures need larger text

---

> ### Author Rebuttal · Authors · 2026-03-31
>
> We sincerely thank Reviewer TP5h for the constructive feedback. We are encouraged that you recognized our MoE integration for handling cyclic demand, highlighted our adaptive anti-hacking reward as a strong practical solution, and found our experimental evaluation thorough and well-structured. We appreciate your efforts in strengthening our work and hope the following addresses your concerns.
>
> >W1: Incremental novelty
>
> We thank the reviewer for this observation! Our primary contribution, rather than proposing a fundamentally new base RL algorithm, is the domain-specific co-design and synthesis of these elements to solve the complex challenges of adaptive delayed matching under non-stationary supply-demand and congestion. Prior RL approaches in this space often struggle with shallow encoders, static travel-time approximations, and fixed reward weights, leading to limited robustness and pathological behaviors. We demonstrate that overcoming these limitations requires addressing the problem holistically through a regime-aware MDP and a specialized MoE-based framework.
>
> Importantly, the components are not used off-the-shelf. The MoE design is adapted to an inherently imbalanced domain, where peak and off-peak regimes should not be forced into uniform expert utilization; our lightweight routing/balancing design is intended to preserve such meaningful specialization. The MFD component is not used as a standalone traffic model, but converted into an efficient travel-time surrogate suitable for millions of RL rollouts. Likewise, the adaptive multiplier is introduced because fixed reward ratios lead to reward hacking in this problem, as shown in Table 1. PPO serves as our primary RL algorithm precisely to demonstrate that the main bottleneck here is not inventing another base RL algorithm, but learning a regime-aware representation that remains stable at city scale.
>
> We emphasize the non-trivial integration and its empirical consequences: a 12M MoE outperforms a 48M dense Transformer, fixed reward weights collapse while the adaptive design remains stable, masking either frequent or rare experts degrades performance, and the same encoder consistently improves multiple RL algorithms (PPO, A2C, ACER, GRPO). We hope this clarifies that the contribution is not simply a collection of known modules, but a unified framework whose components are necessary to make adaptive delayed matching work robustly in practice.
>
> >W2: Single-city evaluation
>
> We completely agree that evaluating on a single city limits claims of generalizability. To address this exact concern, we have conducted new cross-city experiments. We tested both native training and strict zero-shot transfer protocols. Please refer to our response to Reviewer rs6Y W1 & W2 for the complete details.
>
> >W3: Regrading GRPO
>
> We thank the reviewer for pointing out this ambiguity. We want to clarify that GRPO is not a main methodological contribution of this work; PPO serves as our primary training algorithm, and this setup was included solely as an auxiliary baseline to demonstrate that the proposed RAST-MoE encoders' representational benefits generalize across different RL training paradigms. We completely agree that referring to this as a "GRPO variant" is misleading. The contribution lies in the RAST-MoE architecture, regime-aware formulation, and anti-hacking reward, not in modifying RL algorithms. Our implementation does not aim to reproduce canonical GRPO (e.g., critic-free) faithfully, but to introduce a lightweight reward normalization inspired by similar motivations. We will revise the wording to avoid implying a methodological contribution.
>
> Regarding the justification for this specific normalization over simpler alternatives, the tanh-based normalization is used to bound large rewards and stabilize training under highly non-stationary regimes. We will clarify this motivation in the revision.
>
> >W4: Formatting issues, unnumbered equations, and figure clarity
>
> We sincerely apologize for these presentation and formatting oversights. We will carefully reformat the manuscript for the revision.
>
> >Q1: Lack of comparison with existing baselines
>
> Thank you for this question! Qin et al. (2021) is, in fact, included as our main ride-hailing-specific RL baseline. The ACER baseline in Fig. 2 and the Plain MLP + ACER (12M) entry in Appendix Table 2 correspond to this baseline under our data split and simulator. We chose Qin et al. because its formulation is the most similar to our setting among prior ride-hailing RL works, making it the most appropriate task-specific baseline for comparison.
>
> We also matched the parameter count (12M) to our MoE model in order to isolate the benefit of the proposed regime-aware MoE representation under the same problem formulation and comparable model scale. We agree that this point should be stated more explicitly, and we will clarify in the paper that the 12M ACER Plain MLP baseline corresponds to the Qin et al. task-specific baseline.

---

> > ### Author Rebuttal · Reviewer_TP5h · 2026-03-31
> >
> > After carefully considering all the comments and responses, I have raised my score from weak accept to accept.

---

> > > ### Author Response · Authors · 2026-03-31
> > >
> > > Thank you for engaging with our responses and raising the score to 5 (Accept)! We are delighted that our clarifications and new experiments have fully resolved your concerns. Your constructive feedback has been invaluable in helping us strengthen the manuscript. We deeply appreciate your time, effort, and support for our work!

---

### Official Review · Reviewer_rs6Y · 2026-03-11

**Soundness:** 2
**Presentation:** 3
**Significance:** 3
**Originality:** 3
**Overall Recommendation:** 4
**Confidence:** 3

**Summary:**

This paper proposes a regime-aware spatio-temporal MoE framework (RAST-MoE) for DRL in ride-hailing systems. The authors first formulate the problem as an RAST-MDP with an anti-hacking reward design, and then solve it using the proposed RAST-MoE-RL algorithm. Experimental results on a real-world Uber trajectory dataset show that the proposed method can reduce both the average matching delay and pickup delay.

**Compliance With Llm Reviewing Policy:**

Affirmed.

**Final Justification:**

Overall, the two rounds of rebuttal have addressed my primary concerns. After carefully reviewing the paper, the conclusions appear reasonable. Given my limited familiarity with this specific topic, I thus maintain my score at a weak accept.

**Key Questions For Authors:**

1. Most comparisons in the paper focus on different RL algorithms within the proposed framework. Could the authors also compare their method with other existing approaches for ride-hailing platforms to provide a more comprehensive evaluation?
2. The paper mentions unseen demand regimes, but the definition is unclear. What specific demand patterns constitute unseen regimes, and how are these quantified in the experiments?
3. Since the dataset is constructed by the authors, it would be valuable to release both the dataset and the code to facilitate reproducibility and future research.

**Limitations:**

The dataset covers only one city and a single year, which limits the scope of validation. The proposed method should ideally be evaluated on more diverse and large-scale datasets to better assess its robustness and generalization ability.

**Strengths And Weaknesses:**

Strengths.
1. The topic is interesting and practically relevant. The discussion of challenges in adaptive delayed matching provides useful motivation for the proposed approach.
2. The proposed method is efficient yet effective, achieving performance improvements with a relatively lightweight model design.

Weaknesses.
1. The empirical evaluation is limited. Although ride-hailing data can be difficult to collect, evaluating the method on data from only one city within a single year is not sufficiently convincing.
2. The claim of generalization is not well supported. Since all data are collected from a single city, it is unclear whether the method would generalize to different urban environments or demand distributions. Moreover, the dataset split appears to follow a partition within the same data distribution. In standard machine learning practice, validation and test sets should be sufficiently different from the training data to reduce the risk of overfitting to the underlying distribution.

---

> ### Author Rebuttal · Authors · 2026-03-31
>
> We sincerely thank Reviewer rs6Y for the constructive feedback and for recognizing that our formulation of adaptive delayed matching addresses a practically relevant challenge, and that our RAST-MoE-RL approach is an efficient yet effective method achieving strong performance improvements with a relatively lightweight model design. We hope the following addresses your concerns, and we are grateful for your efforts in strengthening our work.
>
> >W1 & W2: Limited empirical evaluation and unsupported generalization claims
>
> To demonstrate generalizability beyond within-city temporal shifts, we conducted cross-city experiments on two new 2019 CPUC dataset cities with distinct topologies: **San Diego City (SD)** (sprawling, low-density, freeway-oriented) and **Alameda County (ALA)** (polycentric, mixing dense cores with sprawling suburbs). These differ from SF in demand distribution, network topology, vehicle density, congestion structure, and spatial demand concentration.
>
> To ensure our test sets are truly out-of-distribution, we used two rigorous protocols:
>
> - **Native Training:** The model is trained and tested on the target city.
> - **Zero-Shot Transfer:** The SF-trained policy is deployed on the target city without fine-tuning. This strict test ensures the agent has never seen the target city's network, zone partition, demand patterns, or congestion profile.
>
> | City | Method | Match Wait (s) | Pickup Wait (s) |
> | --- | --- | --- | --- |
> | **ALA** | Instant Matching | 224 | 892 |
> |  | Zone-Level Congestion | 209 | 832 |
> |  | Baseline RL (ACER Plain MLP 12M) | 209 | 641 |
> |  | **RAST-MoE-RL (SF → ALA)** | **196** | **595** |
> |  | **RAST-MoE-RL (ALA → ALA)** | **191** | **578** |
> | **SD** | Instant Matching | 220 | 947 |
> |  | Zone-Level Congestion | 218 | 873 |
> |  | Baseline RL (ACER Plain MLP 12M) | 215 | 692 |
> |  | **RAST-MoE-RL (SF → SD)** | **205** | **626** |
> |  | **RAST-MoE-RL (SD → SD)** | **208** | **617** |
>
> Addressing W1, method ranking remains consistent across all three distinct urban structures. Native RAST-MoE-RL reduces pickup wait by 30–35% compared to the strongest non-learning heuristic and 7–10% over the baseline RL agent. This confirms our gains stem from the regime-aware MoE representation, not SF-specific overfitting.
>
> Addressing W2, zero-shot transfer results (SF $\to$ ALA/SD) strongly refute distribution memorization. Despite never observing the target cities during training, the SF-trained policy achieves pickup waits within 3% of the native policy in ALA and 1.5% in SD, while outperforming natively trained ACER baselines.
>
> We will include this cross-city evaluation and an expanded train/test split discussion in the revised manuscript.
>
> >Q1: Non-RL approaches for ride-hailing
>
> We agree that comparing against stronger non-RL approaches is essential to justify the RL formulation. Please refer to our response to Reviewer Mp2N, section "W2 & Q1." We implemented three stronger and commonly used congestion- and regime-aware methods, all sharing the identical matching backend and MFD surrogate. RAST-MoE-RL reduces pickup wait by 33% even against the strongest new heuristic.
>
> >Q2: Unclear definition of "unseen demand regimes.”
>
> In our framework, a demand regime is characterized by the joint distribution of zone-level supply--demand imbalance $\bigl(n_p^{(i)}, n_d^{(i)}, \lambda^{(i)}, \mu^{(i)}\bigr)$ and congestion state $v_z^{(h)}$ at a given time. Our experiments test generalization along three axes of increasing distributional shift.
>
> **Temporal regimes (within-city).** Training and test sets use disjoint, non-contiguous 2019 SF time windows (Sec. 6.1.1). The agent encounters unseen temporal demand profiles (e.g., varying weekday/weekend mixes, event surges, seasonal changes). Expert activation heatmaps (Appendix H) show utilization shifting measurably between morning peak (hours 7–10), evening peak (hours 16–19), and off-peak periods, confirming the router identifies distinct regimes and activates different expert combinations than during training.
>
> **Perturbation regimes (within-city).** Appendix I tests out-of-distribution perturbations unseen during training: global shocks (OD travel times scaled by $\eta_h \sim \text{Uniform}(0.8, 1.2)$) and localized incidents (5–12 edges penalized by $\alpha \sim \text{Uniform}(1.5, 3.0)$). Table 7 shows RAST-MoE-RL degrades only 1.1%/1.5% (match/pickup) under global shocks and 6.6%/8.2% under incidents, versus 6.0%/7.0% and 9.5%/20.0% for the ACER baseline, quantifying strict OOD robustness.
>
> **Cross-city regimes (new experiments).** To address the most stringent interpretation of ``unseen regimes,'' we have conducted new experiments on two topologically distinct cities. Those are in the response for your W1W2.
>
> >Q3: Data and code release for reproducibility.
>
> Absolutely! The dataset is not constructed by us. It's from CPUC (link provided in the paper). We will release the code at the camera ready version and researcher can just plug in the CPUC data.

---

> > ### Author Rebuttal · Reviewer_rs6Y · 2026-04-01
> >
> > Thank you for your response and for the newly added experiments on cross-city evaluation, as well as the comparison with non-RL approaches. However, it is somewhat surprising that the matching wait time in the in-domain setting (SD → SD) is worse than in the out-of-domain case (SF → SD). This result appears counterintuitive and would benefit from further explanation.

---

> > > ### Author Response · Authors · 2026-04-03
> > >
> > > Thank you for the follow-up! We appreciate this observation and agree that the result benefits from additional clarification. To clarify, we do not interpret it as showing that the out-of-domain policy (SF $\to$ SD) outperforms the in-domain one (SD $\to$ SD). Rather, the two policies achieve very similar performance and differ only by a small trade-off shift across the two waiting-time metrics.
> > >
> > > This is expected given our objective. RAST-MoE-RL does not optimize a static weighted sum of matching wait and pickup wait. Instead, the reward uses incremental matching/pickup costs with a state-dependent pickup weight $c_p(t)$ and an adaptive Lagrange multiplier $\lambda_t$ that enforces service-quality constraints online. As a result, different policies may converge to slightly different operating points in a similar trade-off region, rather than to a single fixed optimum.
> > >
> > > For SD, the native SD-trained policy trades a marginal increase in matching wait (208s vs. 205s) for a lower pickup wait (617s vs. 626s). This small gap likely reflects a minor difference in release timing. A policy trained directly on SD can become slightly more selective about release timing, which may reduce downstream pickup travel time at the cost of a small extra matching wait. By contrast, the SF-trained zero-shot policy appears to retain a somewhat more aggressive release pattern learned from a denser source city, yielding slightly lower matching wait but higher pickup wait.
> > >
> > > More importantly, the fact that SF $\to$ SD is close to SD $\to$ SD is reasonable. Among our tested target cities, SD appears closer to SF than ALA in terms of demand concentration and congestion structure, whereas ALA is more spatially dispersed. The transfer results reflect this: the SF-trained policy remains near-native on SD, while the gap is slightly larger on ALA. At the same time, the native SD policy still achieves better pickup performance, and both policies remain clearly stronger than the heuristic and plain-MLP RL baselines. You mentioned a very good point, and we think including the explanation in the revision will further strengthen our paper. Thank you!

---

### Official Review · Reviewer_1zNh · 2026-03-11

**Soundness:** 1
**Presentation:** 1
**Significance:** 2
**Originality:** 3
**Overall Recommendation:** 4
**Confidence:** 3

**Summary:**

This paper presents RAST-MoE-RL, a novel reinforcement learning framework designed to tackle the inherent non-stationarity of ride-hailing demand and traffic congestion. At its core, the framework introduces a Regime-Aware Spatio-Temporal Mixture of Experts (RAST-MoE) encoder, which leverages a gating mechanism to dynamically route complex urban features to specialized experts, thereby capturing distinct latent patterns across different traffic regimes (e.g., rush hour vs. off-peak). To overcome the computational bottleneck of traditional microscopic simulators, the authors develop a physics-informed surrogate model based on Macroscopic Fundamental Diagrams (MFD), achieving a balance between physical fidelity and query efficiency. Furthermore, to mitigate the common "reward hacking" issue in RL-based dispatching, the paper employs an adaptive Lagrangian multiplier method to dynamically update reward penalties, ensuring that the pursuit of operational efficiency strictly adheres to Service Level Agreement (SLA) constraints. Experiments conducted on a real-world Uber trajectory dataset from San Francisco demonstrate that the proposed method significantly outperforms standard baselines such as PPO, DQN, and even large-scale dense Transformer architectures in terms of cumulative return, matching latency, and pickup efficiency, while exhibiting superior robustness and parameter efficiency.

**Compliance With Llm Reviewing Policy:**

Affirmed.

**Final Justification:**

The author has resolved my issue, so I have raised my score.

**Key Questions For Authors:**

1. According to the current design, drivers within the same zone receive identical macro-instructions. If a specific area is initially clear, the simultaneous movement of all drivers following a dispatching instruction could instantly trigger localized congestion. This phenomenon might lead to erroneous decision-making based on outdated state information. How does the framework address this potential "herd effect" or synchronized congestion risk?

2. During PPO training, did you implement any routing regularization or load-balancing loss to ensure that all 16 experts undergo sufficient and balanced updates? Without such mechanisms, how do you prevent the model from collapsing into a few dominant experts while leaving others under-trained?

**Limitations:**

1. The references in the manuscript are not sufficiently rigorous. There are instances of duplicated citations (e.g., the paper Drbo), and a entry contain unpolished 'YYYY-MM-DD' placeholders, which do not meet the standards of a professional academic publication.

2. The manuscript does not strictly adhere to the ICML formatting requirements. Specifically, the absence of equation numbers makes it difficult for readers to reference key mathematical derivations and significantly hinders the overall readability of the paper.

3. In the context of ride-hailing dispatching, state transitions are driven not only by a single driver’s decision but are also significantly influenced by the collective actions of all other drivers. Consequently, this task is intrinsically a multi-agent problem. The authors' reliance on a single-agent design may fail to adequately adapt to the competitive and interdependent dynamics of this environment.

4. The performance of the proposed method is validated solely on the San Francisco dataset and its variants.

5. The authors have not provided any reproducible code.

**Strengths And Weaknesses:**

This paper ingeniously introduces the RAST-MoE-RL framework, shifting the ride-hailing dispatching paradigm toward a regime-aware architecture that dynamically specializes via spatio-temporal expert networks. By utilizing a physics-informed surrogate model based on MFDs, it elegantly bridges the gap between high-fidelity traffic dynamics and the computational efficiency required for large-scale RL. Furthermore, the application of an adaptive Lagrangian multiplier method provides a robust mathematical foundation for satisfying SLA constraints, effectively suppressing "reward hacking" behaviors.

However, the manuscript is marred by significant gaps in technical rigor and methodological scope. Critically, the formulation of ride-hailing dispatch as a single-agent problem overlooks the essential multi-agent coordination required to model competitive driver dynamics and state-transition interdependencies. Furthermore, the lack of cross-city validation beyond the San Francisco dataset, coupled with the absence of reproducible code, severely limits the evidence for the framework's generalizability and practical utility. Finally, the submission suffers from poor structural quality, characterized by repeated citations (e.g., Drbo), unpolished 'YYYY-MM-DD' placeholders, and a failure to adhere to ICML formatting standards—specifically the lack of equation numbering—which hinders readability and professional presentation.

---

> ### Author Rebuttal · Authors · 2026-03-31
>
> We sincerely thank Reviewer 1zNh for the constructive feedback and for recognizing our regime-aware RAST-MoE-RL framework, our physics-informed MFD surrogate model, and our adaptive Lagrangian multiplier method that effectively suppresses reward hacking. We hope the following addresses your concerns, and we are grateful for your efforts in strengthening our work.
>
> >Q1: Zone-level actions may cause synchronized congestion
>
> Our zone-level actions do not prescribe a common destination for all drivers in a zone; rather, they dictate *when* vehicles in each zone are released for matching, which is a standard abstraction for tractability in city-scale systems [1,2]. After release, actual assignments are determined by a bipartite matching optimization. Consequently, drivers from the same zone are matched to different passenger requests and dispersed to various pickup locations, naturally mitigating synchronized movement.
>
> Furthermore, any localized congestion is intrinsically handled by our physics-informed MFD surrogate (Section 4.3), which captures density-speed feedback. If a temporary "herd effect" were to occur, the localized density spike would immediately degrade the zone's space-mean speed. Operating in a short-horizon closed loop, this instantly increases pickup costs and lowers matching efficiency, penalizing the agent and correcting the behavior in subsequent decision epochs.
>
> >Q2: Load-balancing and expert collapse prevention during PPO training
>
> This is a good point! Yes. We implement a lightweight load-balancing mechanism (Sec. 5.2) based on learnable bias terms added to the gating logits. This prevents excessive concentration on a few experts while tolerating naturally skewed utilization reflecting real regime frequencies (peak hours are rarer than off-peak).
>
> Empirical evidence confirms no collapse. In Section 6.3, Figure 3 top row shows that all $(E,K)$ configurations maintain broadly distributed activation and there is no setting degenerates to fewer than $K$ active experts. The masking experiments (Figure 3, bottom row) further demonstrate that removing either high-frequency or low-frequency experts causes sharp performance drops; if low-frequency experts were under-trained or dead, masking them would have negligible impact. Appendix H provides additional evidence through 24-hour activation heatmaps: all 16 experts exhibit temporally structured activation patterns corresponding to distinct demand regimes (e.g., peak-onset, off-peak, general), confirming that every expert learns meaningful, non-degenerate behavior under our balancing mechanism.
>
> >L1, L2: Formatting issues
>
> We sincerely apologize for the duplicated citation and the unpolished placeholder. These were oversights during manuscript preparation and will be corrected in the revision. We will add numbering to all key equations in the revised manuscript to improve readability and ease of reference.
>
> >L3: Single-agent formulation may not capture multi-agent dynamics
>
> Thank you for this question. We would like to clarify that our “single-agent” formulation does not model an individual driver making decisions in isolation. The RL agent represents the centralized platform, which observes the global system state and determines the matching-release policy for all zones jointly, following the same modeling paradigm as [1,2]. Therefore, the collective effects of all drivers are not ignored; they are captured in the environment state and transition dynamics through aggregate driver supply, order demand, and the downstream matching outcomes. A multi-agent formulation is typically needed when multiple autonomous entities each learn their own policies. In our setting, however, drivers are not independent learning agents; they are coordinated through the platform-level control and matching mechanism. Thus, a centralized single-agent MDP is an appropriate and standard formulation for this problem.
>
> >L4: Single city concern
>
> We completely agree that evaluating on a single city limits claims of generalizability. To address this exact concern, we have conducted new cross-city experiments. We tested both native training and strict zero-shot transfer protocols. Please refer to our response to Reviewer rs6Y W1 & W2 for the complete details.
>
> >L5: Code availability
>
> We are cleaning our code and will release it at the camera-ready version.
>
> Ref:
>
> [1]Qin et al., 2021. Optimizing matching time intervals for ride-hailing services using RL. Trans. Res. Part C 129, 103239.
>
> [2]Xu et al., 2018. Large-scale order dispatch in on-demand ride-hailing platforms: A learning and planning approach. KDD, 905–913.

---

> > ### Author Rebuttal · Reviewer_1zNh · 2026-04-03
> >
> > Thank you for the your response. The additional cross-city experiments and the further clarification of the zone-level release design have addressed some of my concerns.
> >
> > The response to Q1 is still not fully convincing to me. The authors argue that drivers within the same zone are matched to different passengers and different pickup locations, which would naturally mitigate synchronized movement. In my view, however, this does not rule out the risk of synchronized congestion. In a typical morning peak scenario, even if passengers have different origins and destinations, travel demand remains highly correlated in both time and space, and vehicle flows may still concentrate on similar corridors or hotspot areas, thereby inducing localized congestion. Therefore, I believe this issue still lacks more targeted experimental or analytical support.
> >
> > In addition, the manuscript contains several avoidable  issues, including placeholder text, insufficiently polished references, non-standard equation formatting, and the lack of publicly available code at this stage. Although the authors state that some of these issues will be corrected in the revision, they still negatively affect my assessment of the paper’s overall polish and rigor. For these reasons, I decide to retain my original score.
> >
> > However, since the authors have addressed and clarified several of my previous concerns, I am willing to lower my confidence in the current weak reject decision. If the authors could further provide experimental comparisons with multi-agent approaches, in order to better justify the centralized formulation, I believe this would strengthen the paper and could lead me to raise my score.

---

> > > ### Author Response · Authors · 2026-04-07
> > >
> > > Thank you for your follow-up and for confirming that the cross-city experiments and zone-level release design clarification have addressed some of your concerns! We also appreciate your willingness to consider raising the score with **additional experiments on multi-agent approaches**. This encourages us that the paper is moving in the right direction. Below, we address the remaining points.
> > >
> > > > Synchronized congestion risk.
> > >
> > > We appreciate the reviewer pressing on this point. We offer a mechanistic argument and three levels of empirical evidence.
> > >
> > > Mechanistically, our MFD surrogate (Sec. 4.3) preserves capacity drops at high density. If zone-level release concentrated vehicles on similar corridors, the density spike would immediately degrade space-mean speed, inflating pickup costs. Because the agent operates in a short-horizon closed loop, this penalty is felt within one or two epochs, making corridor-flooding self-correcting during training.
> > >
> > > Empirically: (1) SF itself is among the densest ride-hailing markets in the US, and downtown peaks already exhibit exactly the spatially correlated demand the reviewer describes. Our policy achieves its strongest gains during these peaks, where the MoE router activates more experts to decompose complex congestion patterns. (2) We conducted perturbation experiments strictly harder than usual peaks on the streets: a global shock ($\eta_h \sim \mathrm{Uniform}(0.8,1.2)$) slowing the entire city by up to 20%, and incident shocks injecting 1.5–3× penalties on 5–12 corridor edges—directly simulating corridor flooding. Neither was seen during training. RAST-MoE-RL degrades only 1.1%/1.5% (global) and 6.6%/8.2% (incident), vs. 6.0%/7.0% and 9.5%/20.0% for baseline ACER. A congestion-vulnerable policy could not remain this stable under artificial bottlenecks. (3) Cross-city zero-shot transfers to ALA and SD succeed within 1.5–3% of native performance despite entirely different corridor structures. These results are in Appendices H and I; we will reference them more prominently in the revision.
> > >
> > > > Formatting and code.
> > > >
> > >
> > > We acknowledge these issues. The revision will fix them.
> > >
> > > > **Multi-agent comparison**
> > > >
> > >
> > > We agree that ride-hailing involves interdependent dynamics. Our "single-agent" formulation represents the centralized platform making joint zone-wise decisions, not an isolated driver—collective effects are captured through aggregate supply-demand state and matching outcomes.
> > >
> > > To address the reviewer’s concern more directly, we add a stronger multi-agent comparison with decentralized zone-level execution. Specifically, we treat each zone as an agent, and each agent makes its own binary hold/release decision at execution time from local zone observations together with a small set of shared global summaries. To make this comparison fair, all zone agents share the same actor network, since they solve the same type of local decision problem; however, each agent acts from its own local observation, so execution remains decentralized and agent-specific. We further use a centralized critic trained on the full global state so that each agent receives sufficiently informative training signals under strong inter-zone coupling. The environment, transition dynamics, anti-hacking reward design, action semantics, and evaluation protocol remain unchanged, so this comparison isolates exactly the centralized-vs-decentralized modeling choice.
> > >
> > > To further ensure the comparison is informative, we test both the Transformer encoder and the proposed RAST-MoE encoder under each formulation. The first two rows are the existing results in Section 6.5.
> > >
> > > | Method | Execution type | Matching wait | Pickup wait |
> > > | --- | --- | --- | --- |
> > > | Centralized PPO + Transformer | Centralized | 201 | 569 |
> > > | Centralized PPO + RAST-MoE | Centralized | 182 | 523 |
> > > | Shared-actor multi-agent PPO + Transformer | Decentralized | 217 | 581 |
> > > | Shared-actor multi-agent PPO + RAST-MoE | Decentralized | 204 | 575 |
> > >
> > > Two findings emerge. First, centralized execution consistently outperforms decentralized execution under the same encoder. With RAST-MoE the gap is 22s/52s (match/pickup). This is expected: A centralized agent can jointly reason over which subset of zones to release, whereas decentralized agents cannot fully coordinate this combinatorial decision. Second, RAST-MoE improves both formulations, but the benefit is far larger under centralized execution (pickup drops 46s centralized vs. 6s decentralized), confirming the regime-aware representation is most effective on the complete spatial state.
> > >
> > > These results provide direct empirical support: decentralized agents learn reasonable policies, but the best trade-off requires centralized coordination under strongly coupled dynamics. In the camera-ready version, we plan to allocate another subsection under “6. Experiments” called "6.6 Centralized vs. Decentralized Formulation" to discuss this point. Thank you for helping us strengthen the paper!

---

### Official Review · Reviewer_Mp2N · 2026-03-14

**Soundness:** 3
**Presentation:** 3
**Significance:** 3
**Originality:** 3
**Overall Recommendation:** 4
**Confidence:** 4

**Summary:**

This paper studies adaptive delayed matching in ride-hailing, where the platform must decide whether to match requests immediately or hold them for future batching, so as to balance matching delay and pickup delay under dynamic supply–demand and congestion. The authors formulate this as a regime-aware MDP and propose RAST-MoE-RL, which combines a spatio-temporal Mixture-of-Experts encoder with PPO. Experiments on real-world San Francisco data show improved reward, lower matching delay, and lower pickup delay compared with several RL baselines.

**Compliance With Llm Reviewing Policy:**

Affirmed.

**Final Justification:**

I will keep my original score.

**Key Questions For Authors:**

1. The current empirical comparison is mainly against RL baselines such as PPO, A2C, ACER, DQN, and GRPO-style variants. Since the paper positions adaptive delayed matching as a large-scale operational decision problem in ride-hailing, could the authors also compare against stronger non-RL baselines, such as optimization-based, heuristic batching, or congestion-aware dispatch/matching methods? This would help clarify whether the gains come from the proposed regime-aware RL formulation itself, or mainly from improvements over prior RL implementations.

2. The policy factorizes over zones with independent Bernoulli heads. Given that delayed matching decisions across neighboring zones can be strongly coupled through shared fleet availability and congestion, could the authors discuss whether this factorization limits the ability to model coordinated batching behaviors across zones?

3. The paper reports robustness to unseen demand regimes within the same city and dataset. Could the authors discuss how well the method might generalize to more substantially different operating conditions, such as different cities or more extreme congestion patterns?

**Limitations:**

Yes

**Strengths And Weaknesses:**

### Strengths

1. The paper studies a meaningful and well-formulated sequential decision problem in ride-hailing, and the RAST-MDP formulation explicitly models regime heterogeneity, combinatorial zone-wise actions, adaptive rewards, and congestion-aware travel-time feedback.

2. The method is reasonably well motivated: the MoE encoder is aligned with the regime-dependent nature of the task, and the experiments provide evidence of meaningful expert specialization.

### Weaknesses

1. The method novelty is somewhat limited. The overall optimization remains standard PPO, and the main contribution is more in task formulation, environment design, and representation engineering than in a fundamentally new RL algorithm.

2. The baseline comparison is not fully convincing, since the main experiments are largely against RL baselines; stronger non-RL optimization or dispatch baselines are not thoroughly compared in the core results.

---

> ### Author Rebuttal · Authors · 2026-03-29
>
> We sincerely thank Reviewer Mp2N for the constructive feedback and for recognizing our well-formulated RAST-MDP and well-motivated MoE encoder. We hope the following addresses your concerns, and we are grateful for your efforts in strengthening our work.
>
> > W1. Method novelty is limited / Optimization remains standard PPO
>
> R: We agree that the underlying RL algorithm is PPO. In large-scale, highly non-stationary spatio-temporal control, the fundamental bottleneck is not the RL algorithm, but the MDP formulation and the deep neural network (encoder). Standard networks collapse under drastic regime shifts.
>
> To solve this, our main contribution is the co-design of a physics-informed environment (RAST-MDP) and a regime-aware policy network (RAST-MoE). Integrating MoE into Deep RL is difficult because dynamic data distributions cause standard MoE routing to collapse, while complex task tradeoffs lead to reward hacking. Bridging this gap required two non-trivial, RL-specific innovations:
> 1. Imbalance MoE routing (Sec. 5.2): Standard load-balancing enforces uniform utilization, actively harming performance under skewed regimes (e.g., peak vs. off-peak). Our mechanism enables purposeful, uneven specialization. Fig. 3 masking confirms that all experts are indispensable.
> 2. Anti-hacking Lagrangian reward (Sec. 4.2): Fixed-weight MDPs collapse into pathological strategies (Table 1). Our adaptive multiplier $\lambda$ dynamically enforces service constraints online without brittle, hand-tuned ratios.
>
> As Table 2 demonstrates, equipping A2C, ACER, or GRPO with our RAST-MoE encoder and MDP consistently yields massive gains. We advance the state-of-the-art not by proposing a new RL algorithm, but by proving that specialized, MoE-driven policy networks are the key to unlocking RL for city-scale physical systems.
>
> > W2 & Q1. Need stronger non-RL baselines
>
> We agree that this comparison is essential to justify why RL is necessary. The original submission included two simple heuristics (Instant Matching and Constant-Interval Batch Matching, Table 2 / Appendix C). We have added three substantially stronger congestion- and regime-aware timing-control baselines. To ensure a fair comparison, all share the identical matching backend and MFD surrogate as RAST-MoE-RL:
>
> Fixed-window: sweeps holding time over $W \in$ {0, 20, 40, 60, 90, 120, 180, 240} s, reporting validation-best $W^*$.
>
> Time-of-Day Scheduled: Assigns period-specific $W_r$ to four demand periods in a day.
>
> Zone-Level Congestion-Conditioned: Each zone independently adapts its hold window based on a real-time MFD congestion index $C_t^{(i)} = v_{\text{free}}^{(i)} / v_t^{(i)}$, with tuned thresholds.
> | Method | Match Wait (s) | Pickup Wait (s) |
> |---|---:|---:|
> | Instant Matching | 212 | 825 |
> | Constant-Interval (W=20s) | 201 | 842 |
> | Best Fixed-Window (W*=40s) | 206 | 814 |
> | Time-of-Day Scheduled | 198 | 802 |
> | Zone-Level Congestion-Cond. | 203 | 784 |
> | **RAST-MoE-RL (Ours)** | **182** | **523** |
>
> Even against the strongest congestion-aware heuristic, RAST-MoE-RL reduces pickup wait by **33%** (784s $\to$ 523s). The gap reveals a fundamental limitation of heuristics: they adapt along one axis (time or local congestion), but cannot jointly reason over the full spatio-temporal state at every step like our MoE-enhanced policy.
>
> > Q2. Bernoulli factorization limits cross-zone coordination
>
> We clarify how our framework captures inter-zone dependencies and coordinates batching.
>
> First, while action sampling factorizes as $\pi_\theta(a \mid s) = \prod_{i=1}^N \mathrm{Bernoulli}(a_i \mid \sigma(\ell_i(s)))$, the logits $\ell_i(s)$ are output by a global encoder:
>
> $$h(s) = \mathrm{MoE}_\theta(s), \qquad \ell_i(s) = w_i^\top h(s) + b_i$$
>
> Thus, each zone's probability depends on the entire global spatiotemporal state, ensuring correlated spatial patterns are jointly captured in the logits.
>
> Second, our design is inherently hierarchical. The high-level RL policy selects a subset of zones for matching, $S_1 =$ {$i : a_i = 1$}. The low-level execution then globally pools all passengers and drivers across $S_1$ to solve a single Linear Sum Assignment Problem (LSAP), ignoring zone boundaries. Following Qin (2021), this global batching remedies spatial partitioning tradeoffs and local optimality gaps.
>
> Finally, the LSAP uses a cost matrix $C(t)$ reflecting MFD-informed granular network travel times. This bi-level structure induces coordinated matching outcomes across zones during training.
>
> > Q3. Generalization to unseen regimes and other cities
>
> Please refer to our response to Reviewer rs6Y (W1 & W2), where we present cross-city experiments. The SF-trained policy transfers zero-shot to distinct topologies within 1.5–3% of native performance, confirming the RAST-MoE encoder learns generalizable supply–demand–congestion dynamics.
>
> Ref:
> [1]Cappart, Q., et al., 2023. Machine learning for combinatorial optimization: a methodological tour d’horizon. Eur. J. Oper. Res. 304, 205-228.

---

> > ### Author Rebuttal · Reviewer_Mp2N · 2026-04-03
> >
> > The authors' responses provide more experimental comparison and analysis, but the concern on limited novelty still exists. I will keep my borderline score.

---

> > > ### Author Response · Authors · 2026-04-07
> > >
> > > Thank you for acknowledging our additional experiments and for your positive assessment! We appreciate the reviewer's recognition of the well-formulated RAST-MDP and well-motivated MoE encoder.
> > >
> > > Regarding the novelty concern, we respectfully note that our framework introduces several RL-specific innovations beyond standard PPO, each validated by experiments:
> > >
> > > **Imbalance-aware MoE routing (Sec. 5.2).** Standard MoE load-balancing enforces uniform expert utilization, which actively harms performance when regimes are inherently skewed. Our mechanism enables purposeful specialization. Masking experiments (Fig. 3) confirm all experts are indispensable: removing high-frequency experts degrades pickup wait from 523s to ~780s; removing low-frequency experts causes similar degradation. The 24-hour activation heatmaps (**Appendix H**) show experts specialize into interpretable roles: peak-onset experts, general experts, and off-peak experts. A 12M MoE achieves 182s/523s (match wait/pickup wait), outperforming a 48M dense Transformer at 192s/545s — 4× the parameters yet worse performance (**Table 2, Appendix C**).
> > >
> > > **Anti-hacking Lagrangian reward (Sec. 4.2).** Fixed reward weights consistently produce pathological strategies (**Table 1**): ratio 8:1 collapses matching wait to 0s with 1746s pickup; ratio 1:8 inflates matching to 2236s. Even moderate ratios (2:1, 3:1) yield 278–291s matching vs. our 181s (**Appendix F, Table 5**). Our adaptive multiplier
> > > λ converges stably (**Appendix J, Fig. 7**) and is robust to tolerance
> > > α∈ {2.5%, 5%, 7.5%} with <7% matching variation and <2% pickup variation (**Appendix G, Table 6**). No fixed weight achieves this stability.
> > >
> > > **Physics-informed MFD surrogate (Sec. 4.3).** Under OOD perturbations never seen during training (**Appendix I, Table 7**), RAST-MoE degrades only 1.1%/1.5% (global shock) and 6.6%/8.2% (incident shock), vs. 6.0%/7.0% and 9.5%/20.0% for the ACER baseline, demonstrating the surrogate provides physically grounded training signals.
> > >
> > > **Cross-algorithm and cross-city generality.** The same RAST-MoE encoder improves PPO (182s/523s), GRPO (185s/528s), A2C (197s/555s), and ACER (192s/569s) — all substantially below their MLP counterparts (**Table 2**). Zero-shot SF→ALA achieves 196s/595s vs. native 191s/578s; SF→SD achieves 205s/626s vs. native 208s/617s, both within 1.5–3% of native performance across topologically distinct cities.
> > >
> > > These results collectively demonstrate that the contribution is not a collection of known modules, but a unified framework whose components are each necessary and whose integration produces consistent, large-margin improvements across algorithms, cities, and stress conditions. We hope this merits your continued support. Thank you!

---

### Decision · Program_Chairs · 2026-04-30

**Decision:**

Accept (regular)

**Comment:**

The paper addresses an important and practically relevant ride-hailing problem and presents a well-designed regime-aware RL framework combining a spatio-temporal MoE encoder, congestion-aware surrogate, and adaptive reward design. Although the algorithmic novelty is limited, the rebuttal substantially strengthened the work through stronger heuristic baselines, cross-city transfer, and centralized-versus-decentralized comparisons. These results support that the gains come from the proposed representation and system design rather than tuning alone. With some improvements to presentation and framing in the final version, I recommend acceptance.